# Chemistry and Bioactivity of *Croton* Essential Oils: Literature Survey and *Croton hirtus* from Vietnam

**DOI:** 10.3390/molecules28052361

**Published:** 2023-03-03

**Authors:** Ngoc Anh Luu-dam, Canh Viet Cuong Le, Prabodh Satyal, Thi Mai Hoa Le, Van Huong Bui, Van Hoa Vo, Gia Huy Ngo, Thi Chinh Bui, Huy Hung Nguyen, William N. Setzer

**Affiliations:** 1Vietnam National Museum of Nature, Vietnam Academy of Science and Technology (VAST), No. 18 Hoang Quoc Viet Road, Cau Giay District, Hanoi 100803, Vietnam; 2Vietnam Academy of Science and Technology (VAST), Graduate University of Science and Technology, No. 18 Hoang Quoc Viet Road, Cau Giay District, Hanoi 100803, Vietnam; 3Mientrung Institute for Scientific Research, Vietnam National Museum of Nature, Vietnam Academy of Science and Technology (VAST), 321 Huynh Thuc Khang, Hue 530000, Thua Thien Hue, Vietnam; 4Aromatic Plant Research Center, 230 N 1200 E, Suite 100, Lehi, UT 84043, USA; 5Faculty of Pharmacy, Vinh Medical University, 161 Nguyen Phong Sac, Vinh 461150, Vietnam; 6Department of Pharmacy, Duy Tan University, 03 Quang Trung, Da Nang 550000, Vietnam; 7Center for Advanced Chemistry, Institute of Research and Development, Duy Tan University, 03 Quang Trung, Da Nang 5000, Vietnam; 8Faculty of Biology, University of Education, Hue University, 34 Le Loi St., Hue 530000, Vietnam; 9Department of Chemistry, University of Alabama in Huntsville, Huntsville, AL 35899, USA

**Keywords:** *Croton*, antimicrobial, antiparasitic, larvicidal, molluscicidal

## Abstract

Using essential oils to control vectors, intermediate hosts, and disease-causing microorganisms is a promising approach. The genus *Croton* in the family Euphorbiaceae is a large genus, with many species containing large amounts of essential oils, however, essential oil studies are limited in terms of the number of *Croton* species investigated. In this work, the aerial parts of *C. hirtus* growing wild in Vietnam were collected and analyzed by gas chromatography/mass spectrometry (GC/MS). A total of 141 compounds were identified in *C. hirtus* essential oil, in which sesquiterpenoids dominated, comprising 95.4%, including the main components β-caryophyllene (32.8%), germacrene D (11.6%), β-elemene (9.1%), α-humulene (8.5%), and caryophyllene oxide (5.0%). The essential oil of *C. hirtus* showed very strong biological activities against the larvae of four mosquito species with 24 h LC_50_ values in the range of 15.38–78.27 μg/mL, against *Physella acuta* adults with a 48 h LC_50_ value of 10.09 μg/mL, and against ATCC microorganisms with MIC values in the range of 8–16 μg/mL. In order to provide a comparison with previous works, a literature survey on the chemical composition, mosquito larvicidal, molluscicidal, antiparasitic, and antimicrobial activities of essential oils of *Croton* species was conducted. Seventy-two references (seventy articles and one book) out of a total of two hundred and forty-four references related to the chemical composition and bioactivity of essential oils of *Croton* species were used for this paper. The essential oils of some *Croton* species were characterized by their phenylpropanoid compounds. The experimental results of this research and the survey of the literature showed that *Croton* essential oils have the potential to be used to control mosquito-borne and mollusk-borne diseases, as well as microbial infections. Research on unstudied *Croton* species is needed to search for species with high essential oil contents and excellent biological activities.

## 1. Introduction

In the family Euphorbiaceae, the genus *Croton* has the largest number of species with about 13,000 species, which are distributed mainly in tropical and subtropical areas [1]. Secondary metabolites in this genus include terpenoids, alkaloids, phenolic compounds, and phenylpropanoids [2]. The volatile compounds (essential oils) in *Croton* species are important products with many biological activities such as antioxidant [3,4,5,6,7], antibacterial, antifungal [8,9,10], anti-inflammatory [2,11,12,13], cytotoxic [6,11,14,15,16,17], antitumor [16,17], insecticidal, amebicidal [18], anti-parasitic, anti-ulcerogenic [19,20,21], antinociceptive [22,23], modulation of antibiotic and antifungal activities [24], myorelaxant [25], antispasmodic [26], anxiolytic, [2,27], anthelmintic [28], vasorelaxant [29], and pharmacological effects. 

Mosquito-borne diseases are a global health problem, particularly diseases with a high number of annual infections such as human malaria (148–304 million), dengue (67–136 million), yellow fever (84,000–170,000 in Africa), chikungunya (693,000 in the Americas), Zika (500,000 in the Americas), lymphatic filariasis (31.3–46.7 million), Japanese encephalitis (35,000–50,000), and West Nile fever (2588) [30].

Parasitic diseases transmitted by snails are a serious health problem in addition to mosquito-borne diseases. In 2021, 136 million school-aged children and 115.4 million adults in 51 countries required preventive chemotherapy for schistosomiasis, which represents a slight increase over that in 2020 (239.6 million) [31]. It is estimated that the acute and chronic symptoms of this disease result in a loss of 4.5 million disability-adjusted life years (DALY) [32]. Food-borne trematodiases are most prevalent in east Asia and South America and can result in severe liver and lung disease. Estimates from the WHO show that food-borne trematodes are important causes of disability with an estimated annual total of 200,000 illnesses and more than 7000 deaths per year, resulting in more than 2 million disability-adjusted life years globally [33].

Antibiotic-resistant bacteria are the biggest global health threat today, affecting anyone of any age in any country. The overuse of antibiotics in humans and animals is accelerating this process. Antibiotic resistance leads to an increasing number of infections, less effective treatment with traditional antibiotics, longer hospital stays, higher medical costs, and increased mortality [34]. Overuse and frequent repeated use of traditional insecticides have created resistant mosquito populations [35]. Although resistance to schistosomiasis has not been formally reported, it is a persistent and probable concern [36,37,38,39]. Disease control measures based on a small number of traditional medicines are unsustainable and present risks.

Essential oils, characterized by a complex chemical composition of volatile compounds, are emerging as potential candidates for the control of vectors, intermediate hosts, parasites, and bacteria. The synergistic and antagonistic effects of the complex chemical constituents of essential oils determine the expression of the bioactivity of the essential oil. Many authors believe that it is the complex nature of their chemical composition and the interactions between the components that make it difficult for the target organisms to develop resistance to essential oils [32,40,41]. The trend of synergistic combinations of essential oils with antibiotics [42,43] and pesticides [44,45,46,47,48] to reduce drug use, increase control efficiency, and limit resistance may be an effective solution.

The aim of this study was to obtain and characterize the essential oil from *Croton hirtus* L’Hér from Vietnam and to screen the essential oil for its larvicidal activity against four species of mosquitoes, molluscicidal activity against *Physella acuta*, and antimicrobial activity against a panel of pathogenic microorganisms. In addition, a literature survey of essential oils from *Croton* spp. for the potential control of mosquitoes, snails, parasitic species, and microorganisms was carried out. 

## 2. Results and Discussion

### 2.1. Chemical Composition

The extraction yield of the essential oil was 0.62% (*w*/*w*), which was consistent with the range of 0.3–0.6% mentioned in previous reports. Yields of essential oils from *Croton* spp. species range from 0.02 to 6.41% [49,50].

The chemical composition of *C. hirtus* was dominated by sesquiterpenoids (95.4%). The main chemical components included β-caryophyllene (32.8%), germacrene D (11.6%), β-elemene (9.1%), α-humulene (8.5%), and caryophyllene oxide (5.0%). The full analytical results are available in the Appendix A.

Previous studies have shown that the main chemical composition of this plant’s essential oil varies by season and geographical location. The essential oil samples collected in the study by Simões included spathulenol (26.7%), β-caryophyllene (10.0%), bicyclogermacrene (9.5%), α-cadinol (7.7%), and cubenol (7.0%) [51]. The essential oil samples collected in Teresina showed seasonal variations in the content of major components, namely β-caryophyllene (27.9–37.3%), germacrene D (6.3–33.7%), α-cadinene (7.0–16.1%), δ-cadinene (1.8–13.5%), and α-humulene (3.6–4.6%) [51]. Essential oil samples collected in the Ivory Coast showed main components that included β-caryophyllene (31.75%), germacrene-D (22.57%), and α-humulene (7.42%) [8].

Essential oils from *Croton* species, in addition to being characterized by monoterpenoids and sesquiterpenoids, are particularly characterized by phenylpropanoids [15,28,52,53,54,55,56]. Essential oils of several species are dominated by a major component such as (E)-anethole [52], linalool [57], guaiol, estragole [54], andmethyl eugenol [55].

### 2.2. Larvicidal Activity

The C. hirtus essential oil showed good larvicidal activity against mosquitoes with LC_50_ values in the range of 15.38–78.27 μg/mL (Table 1). The compounds β-caryophyllene, α-humulene, and caryophyllene oxide were active against the larvae of Aedes albopictus with LC_50_ values at 24 h of exposure of 56.87, 43.86, and 20.61 μg/mL, respectively [58]. The compounds germacrene D and β-elemene showed very strong toxicity against the larvae of *Ae. aegypti*, *Ae. albopictus*, and *Cx. quinquefasciatus* [59,60,61,62,63,64].

β-Caryophyllene and its mixtures with caryophyllene oxide and α-humulene (ratio tested relative to the percentage in the essential oil) (Table 2 and Table 3) exhibited distinctly different toxicities against *Ae. aegypti* and *Ae. albopictus*. This mixture played a major role in the larvicidal activity against *Ae. aegypti*, but conversely it did not play a role in the larvicidal activity against *Ae. albopictus*.

### 2.3. Literature Survey

In order to put this current investigation into context, a survey of the literature on *Croton* essential oils and their biological activities was carried out. A total of two hundred and forty-four references were collected, which included one hundred and seven species of *Croton*. The distribution of the study samples collected was as follows: Brazil (one hundred and seventy-six), Venezuela (six), South Africa (six), Cuba (five), Colombia (five), Ecuador (five), Madagascar (four), Nigeria (three), Costa Rica (three), Vietnam (two, not included in this study), Kenya (two), India (two), Cameroon (two), Kenya (two), Peru (two), Ethiopia (two), Mexico (two), Korea (one), Laos (one), Central African Republic (one), Gabon (one), Curacao (one), Benin (one), Argentina (one), Ivory Coast (one), Congo-Brazzaville (one), Sudan (one), Guadeloupe (one), China (one), Thailand (one), Island (one) and Malaysia (one). After reviewing the articles based on the set criteria, seventy-one articles and one book satisfied the criteria and were included herein. Most of the studies collected plant material and were conducted in Brazil.

The previously reported larvicidal activities of *Croton* species against mosquitoes are summarized in Table 4. Most of the essential oils of *Croton* spp. showed good activity (LC_50_ < 100 μg/mL). It is worth noting that *C. zehntneri* essential oil has been characterized by its main component (*E*)-anethole, which has shown very good larvicidal activity against *Ae. aegypti* [52,65]. Furthermore, studies have reported that the yield of essential oil from the leaves of this species is greater than 1%, and it is non-toxic to mice (LD_50_: 3464 mg/kg) [65], so this essential oil may be considered for potential use as a biological pesticide. However, since this essential oil has shown geographical variation in terms of its chemical composition [66], an evaluation of the larvicidal activity of all of its chemotypes has not been performed. 

The main constituents of the essential oils of *Croton* spp. were evaluated for their larvicidal activity against mosquito species (Table 5). There were differences between different authors when reporting the activities of the compounds β-caryophyllene, α-pinene, β-pinene, α-terpineol, α-humulene, and α-phellandrene. This difference may have been due to the different health or developmental stages of the larvae between the groups. The larvicidal activity of (*E*)-anethole was weaker than that of *C. zehntneri* essential oil [52,65], which demonstrated the important role of components in small concentrations, such as anysyl-acetate and dihydroaromadendrene. Bicyclogermacrene did not have synergistic effects with the compounds spathulenol, β-caryophyllene, camphor, or germacrene D in the larvicidal activity of the essential oils of *C. argyrophyllus*, *C. heliotropiifolius,* and *C. pulegiodorus*. This trend was also observed in the larvicidal activity of *Eugenia calycina* essential oil [76].

### 2.4. Molluscicidal and Antiparasitic Activities

The essential oil of *C. hirtus* demonstrated molluscicidal activity against adult *P. acuta* with a 48 h LC_50_ value of 10.09 μg/mL (Table 6). Based on the classification by the World Health Organization (WHO) [114], this essential oil is considered as an active plant molluscicide (LC_50_ < 20 μg/mL). *Croton rudolphianus* essential oil, which is characterized by the main components β-caryophyllene, bicyclogermacrene, δ-cadinene, and germacrene D, was active against *Biomphalaria glabrata* with a 48 h LC_50_ of 47.89 μg/mL [115].

Although only a few studies have been carried out evaluating the toxicity of *Croton* essential oils against snails as disease vectors, a number of essential oil components have been evaluated for their molluscicidal activity [32]. β-Caryophyllene exhibited strong toxicity against *Bulinus truncatus* with an LC_50_ value of 1.66 μg/mL [116].

In contrast to the molluscicidal activity, the antiparasitic activities of essential oils and single components have been extensively studied and have also been studied at the in-vivo level [117,118] (Table 7 and Table 8). Table 7 and Table 8 show that essential oils with high concentrations of β-caryophyllene and/or caryophyllene oxide showed a trend of stronger activity than other essential oils.

### 2.5. Antimicrobial Activity

The essential oil *C. hirtus* exhibited strong antimicrobial activity with MIC values for *E. faecalis* of 8.0 and 16.0 μg/mL for *S. aureus*, *B. cereus*, *E. coli,* and *S. enterica* (Table 9). The compound β-caryophyllene showed strong antimicrobial activity [151], whereas β-elemene in contrast showed weak antimicrobial activity (MIC > 1000 µg/mL) [151]. The essential oils from the leaves and stems of *Orthosiphon stamineus*, which are characterized by the main components β-caryophyllene (24.0–35.1%), α-humulene (14.2–18.4%), and β-elemene (11.1–8.5%), showed strong antimicrobial activity [152]. *Stachys officinalis* essential oil, which is characterized by the main components germacrene D (19.9%), β-caryophyllene (14.1%), and α-humulene (7.5%), exhibited very strong antimicrobial activity [153]. The main components showed weaker antimicrobial activity than essential oils themselves, which suggests synergistic effects between them.

The antimicrobial activities of the essential oils of *Croton* species are summarized in Table 10. It is noteworthy that there are reports on the synergism of essential oils and antibiotics, although essential oils alone or antibiotics alone have shown weakness.

## 3. Materials and Methods

### 3.1. Plant Material and Isolation of Essential Oil

The specimens of *Croton hirtus* L’Hér. were collected at Phong Dien Nature Reserve, Thua Thien Hue Province (16°24′15,84″ N 107°12′00,01″ E, 415 m elevation) in July 2022. The specimen (label: NCXS-H 110) of this species was identified by Van Huong Bui and was deposited in the Vietnam National Museum of Nature (VNMN) herbarium. The fresh aerial parts were chopped and hydrodistilled with a Clevenger apparatus (Witeg Labortechnik, Wertheim, Germany) for 6 h. The EO was dried over anhydrous Na_2_SO_4_ and stored at 4 °C until use.

### 3.2. Gas Chromatographic Analysis

Gas chromatography–mass spectral analyses (GC–MS) of *C. hirtus* essential oil were carried out using previously published instrumentation and protocols [58,114]. A Shimadzu GCMS-QP2010 Ultra (Shimadzu Scientific Instruments, Columbia, MD, USA) with a ZB-5 ms fused silica capillary column (60 m length, 0.25 mm diameter, and 0.25 μm film thickness) (Phenomenex, Torrance, CA, USA), He carrier gas, 2.0 mL/min flow rate, injection and ion source temperatures of 260 °C, and a GC oven program of 50 to 260 °C at 2.0 °C/min was used. A 0.1 μL amount of a 5% (*w*/*v*) sample of essential oil in CH_2_Cl_2_ was injected in split mode with a 24.5:1 split ratio. Identification of the essential oil components was carried out with a comparison of MS fragmentation and retention indices (RI) with those available in the databases [189,190,191,192]. Quantification was performed using external standards of representative compounds from each compound class.

### 3.3. Screening for Larvicidal Activity

Two species of *Aedes* mosquitoes were maintained continuously at Duy Tan University [193]. Egg rafts of *Cx. fuscocephala* were collected from rice fields in Hoa Vang district, Da Nang (GPS: 16°00′49″ N, 108°06′12″ E). Egg rafts of *Cx. quinquefasciatus* were collected from car tires containing decomposing organic matter in Da Nang City. Each *Culex* egg raft was hatched separately in plastic trays with tap water overnight to facilitate the precise examination of the species. The larvae were fed a mixture of dog food and yeast (ratio 3:1, *w*/*w*).

Essential oil and purified compounds were dissolved with ethanol (Sigma–Aldrich, Ho Chi Minh, Vietnam) to obtain a 1% stock solution. Twenty-five larvae of each mosquito species were transferred into 250 mL beakers containing 150 mL of distilled water. Different volumes of each the stock solution were transferred into the beakers containing larvae to obtain exposure concentrations of 100, 75, 50, 25, 12.5, and 6.25 μg/mL. Each concentration was cloned 4 times, and permethrin (Sigma–Aldrich) was used as a positive control. After 24 h and 48 h of exposure, the larvae were determined for mortality.

### 3.4. Screening for Molluscicidal Activity

Adult snails about 1.0 cm in size were collected in the wild (GPS: 16°01′08″ N, 108°07′44″ E), and they were acclimated to laboratory conditions 24 h prior to testing. The five snail adults were transferred into 200 mL plastic beakers, which were then filled with 195 mL of distilled water. The adults were exposed to concentrations of 50, 25, 12.5, 6.25, and 3.125 μg/mL. After 24 h of exposure, the snails were recovered by transferring them to plastic beakers containing only distilled water. After 24 h of recovery, the number of dead snails at the exposure concentrations was recorded. Copper sulfate (Xilong Chemicals, Shantou, China) was used as a positive control.

### 3.5. Screening for Antimicrobial Activity

The ATCC international standard for control of microorganisms include three Gram-negative bacteria strains (*E. coli* ATCC25922, *P. aeruginosa* ATCC27853, and *S. enterica* ATCC13076), three Gram-positive strains (*E. faecalis* ATCC299212, *S. aureus* ATCC25923, and *B. cereus* ATCC 14579), and a strain of *C. albicans* ATCC10231, which were provided by the National Institute for Food Control, Vietnam.

The antimicrobial activity was analyzed based on the multi-concentration dilution method. Samples of essential oils or pure compounds were diluted in DMSO at a decreasing concentration range of 256, 128, 64, 32, 16, 4, and 2 µg/mL, with three replicates for each concentration. Microbial solutions were prepared at a concentration of 2 × 10^5^ CFU/mL, and antimicrobial assays were carried out in 96-well microtiter plates. A 5.12 μL sample solution with a 10 mg/ml concentration was aspirated into the first row containing 100 μL of LB medium and then diluted successively by concentration into rows containing 50 μL until reaching a concentration of 2 μg/mL. Then, 50 μL of microbial solution was added at a concentration of 2 × 10^5^ CFU/ml and incubated at 37 °C. After incubation for 24 hours at 37 °C, the absorbance at 650 nm was measured using a microplate reader (Epoch, BioTek Instruments Inc., Winooski, VT, USA) [194]. Streptomycin, kanamycin, tetracycline, nystatin, and cycloheximide (all compounds were purchased from Sigma–Aldrich) were used as positive controls.

### 3.6. Data Analysis

Mortality data were analyzed by log-probit analysis [195] to acquire LC_50_ and LC_90_ values as well as 95% confidence limits using Minitab^®^ version 19.2020.1 (Minitab, LLC, State College, PA, USA).

### 3.7. Literature Survey

The materials used in this literature survey were searched on the databases https://scholar.google.com, https://pubmed.ncbi.nlm.nih.gov, and https://www.researchgate.net (accessed on 1 January 2023) with a keyword structure including “essential oil” and *Croton*; bioactive keywords that were searched for included antimicrobial, antifungal, antibacterial, antiparasitic, and “mosquito larvicidal”. There were no language restrictions for the selection of articles. The following criteria were included when considering articles:(1)The articles fully reported the chemical composition and mosquito larvicidal activity of the essential oils.(2)The articles fully reported the chemical composition and the molluscicidal and antiparasitic activities of the essential oils.(3)The articles fully reported the chemical composition and antibacterial activity of the essential oils.(4)Articles that reported unreliable GC/MS analysis results, such as chemical compositions that did not match the elution order, retention time, and retention index, were not considered.

## 4. Conclusions

This work presented a literature survey of the volatile phytochemistry and biological activities of *Croton* species and illustrated the potential utility of these essential oils. Furthermore, the essential oil composition, mosquito larvicidal, molluscicidal, and antimicrobial activities of *Croton hirtus* from Vietnam was included, which adds to our knowledge of the genus *Croton*. β-Caryophyllene occurred abundantly in the essential oils [196], and it was present in most of the essential oils of the *Croton* species. Mixtures of β-caryophyllene, α-humulene, and caryophyllene oxide showed synergistic or antagonistic effects against various organisms. Investigations into the bioactivity of combinations of β-caryophyllene with other major compounds in their respective percentages in the essential oils will help in the development of β-caryophyllene-based products. Based on the results of the antiparasitic activity of β-caryophyllene, caryophyllene oxide, and the *Croton* essential oils containing these two compounds, we suggest that an investigation into the antiparasitic activity of *C. hirtus* essential oil may provide interesting results.

## Figures and Tables

**Table 1 molecules-28-02361-t001:** Mosquito larvicidal activity of *Croton hirtus* essential oil (μg/mL).

Test Organism	24-h LC_50_ (95% Limits)	24-h LC_90_ (95% Limits)	48-h LC_50_ (95% Limits)	48-h LC_90_ (95% Limits)
*Aedes aegypti* (third–fourth instar)	29.71 (28.04–31.85)	39.55 (36.11–45.53)	25.67 (24.08–27.82)	34.59 (31.57–39.82)
*Aedes albopictus* (third–fourth instar)	15.38 (14.51–16.53)	20.10 (18.38–22.99)	14.29 (13.31–15.55)	19.36 (17.61–22.31)
*Aedes albopictus* (fourth instar, wild)	78.27 (71.15–86.89)	128.37 (115.56–146.48)	60.0 (53.17–67.73)	155.80 (129.95–197.41)
*Culex quinquefasciatus* (third instar)	50.84 (45.90–56.15)	100.87 (88.39–119.70)	30.42 (28.37–33.78)	38.98 (34.82–48.06)
*Culex fuscocephala* (third–fourth instar)	65.84 (62.24–69.56)	95.15 (89.61–102.41)	38.21 (33.34–43.62)	156.96 (123.03–220.31)

**Table 2 molecules-28-02361-t002:** Larvicidal activity of major compounds against *Aedes aegypti* (μg/mL).

Compounds	LC_50_ (95% Limits)	LC_90_ (95% Limits)	χ^2^	*p*
	24 h			
Caryophyllene oxide	39.65 (35.83–42.53)	49.41 (46.31–53.36)	0.011	1.00
α-Humulene	48.19 (44.33–52.29)	87.64 (78.81–100.02)	1.890	0.596
β-Caryophyllene	111.66 (105.55–118.0)	160.10 (151.39–170.85)	3.782	0.436
Caryophyllene oxide/α-Humulene/β-Caryophyllene: 7:2:1	9.54 (8.06–11.00)	23.08 (20.48–26.83)	11.905	0.018
	48 h			
Caryophyllene oxide	37.92 (34.73–40.82)	47.94 (44.58–52.34)	0.015	1.00
α-Humulene	36.22 (33.15–39.51)	70.58 (62.82–81.67)	5.124	0.163
β-Caryophyllene	94.43 (88.37–100.84)	145.91 (136.85–157.04)	1.821	0.769
Caryophyllene oxide/α-Humulene/β-Caryophyllene: 7:2:1	8.97 (7.48–10.40)	22.30 (19.78–25.93)	9.252	0.055

**Table 3 molecules-28-02361-t003:** Larvicidal activity of major compounds against *Aedes albopictus* (μg/mL).

Componds	LC_50_ (95% Limits)	LC_90_ (95% Limits)	χ^2^	*p*
	24 h			
Caryophyllene oxide	38.68 (35.84–41.44)	53.28 (49.31–58.93)	0.212	0.976
α-Humulene	31.49 (28.62–34.67)	65.14 (56.73–78.08)	8.186	0.042
β-Caryophyllene	30.11 (27.65–32.81)	53.88 (47.80–63.20)	1.865	0.601
Caryophyllene oxide/α-Humulene/β-Caryophyllene: 7:2:1	>50	>50	Nd	Nd
	48 h			
Caryophyllene oxide	33.95 (31.55–36.61)	49.37 (44.93–56.02)	2.136	0.545
α-Humulene	26.44 (24.0–29.13)	55.80 (48.53–66.95)	5.662	0.129
β-Caryophyllene	25.70 (23.46–28.17)	50.26 (44.15–59.60)	3.258	0.354
Caryophyllene oxide/α-Humulene/β-Caryophyllene: 7:2:1	>50	>50	Nd	Nd

Nd: not determined.

**Table 4 molecules-28-02361-t004:** Summary of *Aedes aegypti* larvicidal activity of essential oils of *Croton* spp.

Species	Yield (**%**)	Main Components ^a^	24-h LC_50_ (μg/mL)	24-h LC_90_ (μg/mL)	Ref.
*Croton argyrophylloides* Muell	Aerial parts: Nd	*trans*-β-Guaiene, α-pinene, β-elemene, 1,8-cineole.	94.6	Nd	[65]
*Croton argyrophyllus* Kunth	Dried leaves: 0.48	Spathulenol, β-caryophyllene, α-pinene, bicyclogermacrene.	310	700	[67]
*Croton heliotropiifolius* Kunth	Dried leaves: 0.2	β-Caryophyllene, bicyclogermacrene, germacrene D.	544	Nd	[68]
*Croton jacobinenesis* Baill.	Leaves: 0.80	1,8-Cineole, β-caryophyllene, viridiflorene, α-pinene, β-pinene.	79.3	Nd	[69]
	Stalks: 0.70	δ-Cadinene, β-caryophyllene, γ-muurolene, γ-cadinene, 6,9-guaiadiene, viridiflorene,	117.2	Nd	[69]
	Inflorescences: 0.05	1,8-Cineole, β-caryophyllene, viridiflorene, α-pinene.	65.8	Nd	[69]
*Croton linearis* Jacq	Fresh leaves: 1.50	1,8-Cineole, sabinene, 10-*epi*-γ-eudesmol, hinesol	64.24	143.85	[70]
*Croton nepetaefolius* Bail	Aerial parts: Nd	Methyleugenol, α-copaene, croweacin, caryophyllene oxide.	66.4	154	[65]
*Croton pulegiodorus* Baill.	Dried leaves: 5.0	β-Caryophyllene, bicyclogermacrene, germacrene D.	159	Nd	[68]
*Croton regelianus* Müll. Arg.	Fresh leaves: 1.3	*p*-Cymene, ascaridole, camphor, α- phellandrene.	66.74	Nd	[71]
	Fresh leaves: 0.5	Ascaridole, *p*-cymene, α-terpinene, γ-terpinene.	24.22	Nd	[71]
*Croton rhamnifolioides* Pax and K. Hoffm.	Fresh leaves: Nd	Sesquicineole, α-phellandrene, β-caryophyllene, 1,8-cineole,	122.3	Nd	[72]
	Dried leaves: 0.80%	1,8-Cineole, *o*-cymene, α-pinene, α-phellandrene, sabinene.	89.0	Nd	[72]
*Croton sonderianus* Muell	Aerial parts: Nd	Spathulenol, β-caryophyllene, caryophyllene oxide, 1,8-cineole	54.5	Nd	[65]
	Leaves: Nd	β-Phellandrene, *trans*-β-guaiene, α-pinene, β-caryophyllene, γ-muurolene.	104	119	[73]
*Croton tetradenius* Baill.	2.73	Camphor, γ-terpineol, α-terpinene, *p*-cymene, γ-terpinene.	152	297	[74]
Blend (1:1, *w*/*w*) of *Croton argyrophyllus* Kunth. and *Croton tetradenius* Baill.	Nd	Camphor, isopinocampheol, β-caryophyllene, spathulenol.	160	400	[75]
*Croton zehntneri* Pax et Hoffm	Aerial parts: Nd	(*E*)-Anethole	26.2	Nd	[65]
	Aerial parts: Nd	(*E*)-Anethole	28	32	[73]
*Croton zehntneri* Pax et Hoffm	Leaves: 1.04	(*E*)-Anethole	56.2	Nd	[52]
	Stalks: 0.46	(*E*)-Anethole, *p*-anisaldehyde, anisyl acetate, estragole.	51.3	Nd	[52]
	Inflorescences: 0.30	(*E*)-Anethole	57.5	Nd	[52]
	Leaves: Nd	α-Pinene, *trans*-β-guaiene, β-pinene, β-gurjunene, β-elemene.	102	129	[73]
	Leaves: Nd	Methyleugenol, α-copaene, β-caryophyllene	84	Nd	[73]

^a^: The order of the compounds is sorted by percentage from high to low and greater than 5.0%. Nd = not determined.

**Table 5 molecules-28-02361-t005:** Summary of mosquito larvicidal activity of main components in *Croton* spp. essential oils (24 h or 48 h exposure).

Compound	LC_50_, μg/mL	LC_90_ μg/mL	Mosquito	Ref.
(*E*)-Anethole	69.2	Nd	*Aedes aegypti*	[52]
	50.19	65.21	*Aedes aegypti*	[77]
	34.41–38.98	71.03–82.72	*Aedes aegypti*	[78]
	67.1–85.5	Nd	*Aedes aegypti*	[79]
	42	>50	*Aedes aegypti*	[80]
	50 < LC_50_ < 100	Nd	*Aedes albopictus*	[81]
	73.99	109.86	*Ochlerotatus caspius*	[82]
	24.8 μL/L	32.1 μL/L	*Culex quinquefasciatus*	[83]
	21	34	*Culex quinquefasciatus*	[84]
	16.56	25.29	*Culex pipiens*	[85]
Ascaridole (89.5%)	41.85	74.45	*Culex quinquefasciatus*	[86]
	9.60	Nd	*Aedes aegypti*	[71]
α-Asarone	22.38–23.82	Nd	*Culex pipiens pallens*	[87]
Bicyclogermacrene	11.1	22.14	*Aedes albopictus*	[62]
	12.5	24.2	*Culex tritaeniorhynchus*	[62]
	10.3	20.9	*Anopheles subpictus*	[62]
Borneol	>500	>500	*Aedes aegypti*	[79]
(+)-Borneol	>100	Nd	*Culex pipiens pallens*	[88]
(−)-Borneol	>100	Nd	*Culex pipiens pallens*	[88]
δ-Cadinene	8.23	Nd	*Anopheles stephensi*	[60]
	9.03	Nd	*Aedes aegypti*	[60]
	9.86	Nd	*Culex quinquefasciatus*	[60]
Camphor	>250	>250	*Culex quinquefasciatus*	[84]
	129.17	192.42	*Anopheles anthropophagus*	[89]
	>500	Nd	*Aedes aegypti*	[79]
	>50	>50	*Aedes aegypti*	[80]
1,8-Cineole	>100	Nd	*Aedes aegypti*	[72]
	57.2	Nd	*Aedes aegypti*	[90]
	1381	Nd	*Aedes aegypti*	[91]
	53.63	Nd	*Aedes aegypti*	[92]
	>100	Nd	*Aedes aegypti*	[93]
	>100	>100	*Culex pipiens pallens*	[88]
	191	207	*Culex pipiens molestus*	[94]
	>200	Nd	*Culex pipiens*	[95]
	>50.0	>50.0	*Aedes aegypti*	[96]
	>50.0	>50.0	*Aedes albopictus*	[96]
	>100	Nd	*Aedes albopictus*	[81]
	>200	>200	*Aedes albopictus*	[97]
	>250	>250	*Culex quinquefasciatus*	[84]
β-Caryophyllene	1038	Nd	*Aedes aegypti*	[68]
	136.85	280.86	*Aedes aegypti*	[98]
	298.4	1227.3	*Aedes aegypti*	[99]
	1202	Nd	*Aedes aegypti*	[91]
	>50	>50	*Aedes aegypti*	[80]
	>50	>50	*Aedes aegypti*	[100]
	29.97	48.34	*Aedes aegypti*	[61]
	>100	Nd	*Aedes aegypti*	[93]
	54.95	Nd	*Aedes aegypti*	[101]
	73.4	434.22	*Aedes albopictus*	[99]
	44.8	Nd	*Aedes albopictus*	[102]
	53.14	Nd	*Aedes albopictus*	[101]
	>200	>200	*Aedes albopictus*	[97]
	>100	Nd	*Aedes albopictus*	[81]
	31.09	54.92	*Aedes albopictus*	[61]
	>50	>50	*Aedes albopictus*	[100]
	69.60	164.59	*Culex quinquefasciatus*	[98]
	165.4	220.6	*Culex quinquefasciatus*	[103]
	44.99	Nd	*Culex pipiens pallens*	[101]
	48.2	Nd	*Culex tritaeniorhynchus*	[102]
	>200	Nd	*Anopheles anthropophagus*	[89]
	134.77	Nd	*Anopheles sinensis*	[104]
	41.7	Nd	*Anopheles subpictus*	[102]
	28.86	51.82	*Anopheles nuneztovari*	[61]
	26.52	46.51	*Anopheles triannulatus*	[61]
	25.14	54.73	*Anopheles darlingi*	[61]
	26.36	53.92	*Anopheles albitarsis*	[61]
	60.17	Nd	*Anopheles sinensis*	[101]
Caryophyllene oxide	49.46	115.38	*Anopheles anthropophagus*	[89]
	39.09	Nd	*Anopheles sinensis*	[104]
	125	Nd	*Aedes aegypti*	[91]
	>50	>50	*Aedes aegypti*	[80]
	>50	>50	*Aedes aegypti*	[100]
	29.8 (1 day old)	74.1 (1 day old)	*Aedes aegypti*	[105]
	20.61	27.56	*Aedes albopictus*	[58]
	>50	>50	*Aedes albopictus*	[100]
	98.52	144.5	*Culex quinquefasciatus*	[58]
*p*-Cymene	19.2	41.3	*Aedes aegypti*	[96]
	21.86–49.25	55.02–115.51	*Aedes aegypti*	[78]
	17.05	27.30	*Aedes aegypti*	[98]
	43.3	>50.0	*Aedes aegypti*	[100]
	69.4	95.2	*Aedes aegypti*	[106]
	37.1	>100.0	*Aedes aegypti*	[107]
	12.49	Nd	*Aedes aegypti*	[92]
	36.9 (1 day old larvae)	54.4 (1 day old larvae)	*Aedes aegypti*	[108]
	23.3 (1 day old larvae)	46.7 (1 day old larvae)	*Aedes aegypti*	[109]
	>500	Nd	*Aedes aegypti*	[79]
	25 < LC_50_ < 50	<50	*Aedes aegypti*	[93]
	33.93	Nd	*Aedes aegypti*	[101]
	46.7	>50.0	*Aedes albopictus*	[96]
	34.9	>50.0	*Aedes albopictus*	[100]
	25.9	66.3	*Aedes albopictus*	[107]
	35.10	Nd	*Aedes albopictus*	[101]
	68.3	95.0	*Aedes albopictus*	[106]
	19.4	28.8	*Aedes albopictus*	[97]
	50 < LC_50_ < 100	Nd	*Aedes albopictus*	[81]
	21	30	*Culex quinquefasciatus*	[84]
	15.13	25.41	*Culex quinquefasciatus*	[98]
	29.34	Nd	*Culex pipiens pallens*	[101]
	38.07	Nd	*Anopheles sinensis*	[101]
β-Elemene	10.26	20.02	*Anopheles subpictus*	[63]
	11.15	21.32	*Aedes albopictus*	[63]
	12.05	22.40	*Culex tritaeniorhynchus*	[63]
Elemicin	>100	Nd	*Aedes albopictus*	[81]
Estragole	14.01	24.41	*Culex quinquefasciatus*	[60]
	12.70	22.32	*Aedes aegypti*	[60]
	11.01	19.79	*Anopheles stephensi*	[60]
	38.56	95.90	*Anopheles anthropophagus*	[110]
	41.67	107.89	*Anopheles sinensis*	[110]
Eugenol	7.53	12.35	*Ochlerotatus caspius*	[82]
	117	180	*Culex quinquefasciatus*	[84]
	82.2–142.9	Nd	*Aedes aegypti*	[79]
	12.5 < LC_50_ < 25	50 < LC_90_ < 100	*Aedes albopictus*	[81]
Eugenol (74.0%)	18.28	43.11	*Culex pipiens*	[85]
Germacrene D	49.81	106.19	*Anopheles anthropophagus*	[89]
	59.5	96.4	*Anopheles stephensi*	[59]
	63.6	100.7	*Aedes aegypti*	[59]
	21.28	37.04	*Culex quinquefasciatus*	[60]
	18.76	33.37	*Aedes aegypti*	[60]
	35.96	61.46	*Aedes aegypti*	[61]
	33.51	66.43	*Aedes albopictus*	[61]
	16.95	30.95	*Anopheles stephensi*	[60]
	32.36	58.68	*Anopheles nuneztovari*	[61]
	30.31	58.53	*Anopheles triannulatus*	[61]
	24.49	45.11	*Anopheles darlingi*	[61]
	31.22	55.46	*Anopheles albitarsis*	[61]
α-Humulene	37.89	83.95	*Aedes aegypti*	[58]
	53.05	82.78	*Aedes aegypti*	[103]
	28.11	51.1	*Aedes aegypti*	[61]
	>100	Nd	*Aedes aegypti*	[93]
	108.06	Nd	*Aedes aegypti*	[101]
	38.72	63.40	*Aedes albopictus*	[58]
	106.25	Nd	*Aedes albopictus*	[101]
	6.86	12.98	*Aedes albopictus*	[63]
	28.89	48.28	*Aedes albopictus*	[61]
	87.81	140.0	*Culex quinquefasciatus*	[58]
	108.3	158.2	*Culex quinquefasciatus*	[103]
	96.35	Nd	*Culex pipiens pallens*	[101]
	7.39	13.68	*Culex tritaeniorhynchus*	[63]
	107.35	Nd	*Anopheles sinensis*	[101]
	6.19	12.03	*Anopheles subpictus*	[63]
	26.63	49.56	*Anopheles nuneztovari*	[61]
	33.08	61.41	*Anopheles triannulatus*	[61]
	30.36	68.88	*Anopheles darlingi*	[61]
	37.42	82.58	*Anopheles albitarsis*	[61]
*R*-(+)-limonene	11.88	17.78	*Aedes aegypti*	[77]
	37	Nd	*Aedes aegypti*	[91]
	25 < LC_50_ < 50	LC_90_ < 100	*Aedes albopictus*	[81]
	71.9	96.9	*Aedes aegypti*	[106]
	25 < LC_50_ < 50	50 < LC_90_ < 100	*Aedes aegypti*	[93]
	41.2	88.2	*Aedes albopictus*	[106]
(±)-Limonene	17.04	Nd	*Aedes aegypti*	[101]
	14.05	Nd	*Culex pipiens pallens*	[101]
*S*-(−)-Limonene	25 < LC_50_ < 50	LC_90_ < 100	*Aedes albopictus*	[81]
	25 < LC_50_ < 50	25 < LC_90_ < 50	*Aedes aegypti*	[93]
	29.1 (1 day old larvae)	81.3 (1 day old larvae)	*Aedes aegypti*	[105]
Limonene	18.1	41.0	*Aedes aegypti*	[96]
	19.4	>50.0	*Aedes aegypti*	[100]
	32.7	50.0	*Aedes albopictus*	[96]
	31.63	41.51	*Culex quinquefasciatus*	[103]
	15.0	34.0	*Aedes albopictus*	[100]
Linalool	155.73	237.29	*Ochlerotatus caspius*	[82]
	>50.0	>50.0	*Aedes aegypti*	[100]
	>50.0	>50.0	*Aedes albopictus*	[100]
	>500	Nd	*Aedes aegypti*	[79]
	>100	Nd	*Aedes aegypti*	[93]
	38.64	69.08	*Aedes aegypti*	[60]
	35.17	63.45	*Anopheles stephensi*	[60]
	42.28	73.13	*Culex quinquefasciatus*	[60]
(−)-Linalool	169.6	220.5	*Aedes albopictus*	[97]
Methyleugenol	36.5 (1 day old larvae)	99.2 (1 day old larvae)	*Aedes aegypti*	[109]
	12.5 < LC_50_ < 25	Nd	*Aedes albopictus*	[111]
	53.30–67.02	Nd	*Culex pipiens pallens*	[87]
β-Myrcene	>100	Nd	*Aedes albopictus*	[81]
	167	218	*Culex quinquefasciatus*	[84]
	27.9	Nd	*Aedes aegypti*	[102]
	>500	Nd	*Aedes aegypti*	[79]
	23.5	Nd	*Aedes albopictus*	[102]
β-Myrcene	35.8	>100.0	*Aedes aegypti*	[107]
	27.0	75.4	*Aedes albopictus*	[107]
	35.8	>100.0	*Aedes aegypti*	[106]
	>100.0	Nd	*Aedes aegypti*	[93]
	27.0	75.5	*Aedes albopictus*	[106]
α-Phellandrene	39.3	Nd	*Aedes aegypti*	[72]
	16.6	36.9	*Aedes aegypti*	[96]
	39.9	>50.0	*Aedes albopictus*	[96]
	25 < LC_50_ < 50	<100	*Aedes albopictus*	[81]
	>100	Nd	*Aedes aegypti*	[93]
	39.3	Nd	*Aedes aegypti*	[72]
α-Pinene	>50.0	>50.0	*Aedes aegypti*	[96]
	79.1	>100.0	*Aedes aegypti*	[107]
	45.17–45.70	92.52–96.49	*Aedes aegypti*	[78]
	15.4	Nd	*Aedes aegypti*	[90]
	>100.0	>100.0	*Aedes aegypti*	[106]
	15.87	Nd	*Aedes aegypti*	[92]
	>500	Nd	*Aedes aegypti*	[79]
	>100	Nd	*Aedes aegypti*	[93]
	>50.0	>50.0	*Aedes albopictus*	[96]
	74.0	>100.0	*Aedes albopictus*	[107]
	80.6	>100.0	*Aedes albopictus*	[106]
	74.0	>100.0	*Aedes albopictus*	[106]
	>100.0	Nd	*Aedes albopictus*	[81]
	68.68–72.30	113.88–114.43	*Aedes albopictus*	[112]
	95	581	*Culex quinquefasciatus*	[84]
	58.44–61.46	124.2–144.56	*Culex pipiens*	[113]
(1*R*)-(+)-α-Pinene	47	62	*Culex pipiens molestus*	[94]
	>100	Nd	*Aedes albopictus*	[111]
(1*S*)-(−)-α-Pinene	49	85	*Culex pipiens molestus*	[94]
	>100	Nd	*Aedes albopictus*	[111]
β-Pinene	65	359	*Culex quinquefasciatus*	[84]
	32.97–35.13	93.11–105.59	*Aedes aegypti*	[78]
	12.1	Nd	*Aedes aegypti*	[90]
	23.63	32.12	*Aedes aegypti*	[103]
	>500	Nd	*Aedes aegypti*	[79]
	27.69	49.91	*Aedes aegypti*	[60]
	50 < LC_50_ < 100	Nd	*Aedes aegypti*	[93]
	>100	Nd	*Aedes albopictus*	[81]
	42.39–47.33	63.10–73.11	*Aedes albopictus*	[112]
	30.46	41.58	*Culex quinquefasciatus*	[103]
	32.23	56.58	*Culex quinquefasciatus*	[60]
	36.53–66.52	76.27–109.53	*Culex pipiens*	[113]
	32.2	Nd	*Anopheles stephensi*	[102]
	23.17	43.39	*Anopheles stephensi*	[60]
Sabinene	74.1	>100.0	*Aedes aegypti*	[106]
	21.20	39.22	*Aedes aegypti*	[60]
	39.5	71.4	*Aedes albopictus*	[106]
	6.25 < LC_50_ < 12.5	25 < LC_90_ < 50	*Aedes albopictus*	[81]
	25.01	45.15	*Culex quinquefasciatus*	[60]
	19.67	36.45	*Anopheles stephensi*	[60]
Spathulenol	>100	Nd	*Aedes aegypti*	[76]
γ-Terpinene	30.7	>50.0	*Aedes aegypti*	[96]
	26.8	68.7	*Aedes aegypti*	[107]
	9.76	16.99	*Aedes aegypti*	[77]
	11.25	21.55	*Aedes aegypti*	[98]
	24.58–44.80	72.55–100.71	*Aedes aegypti*	[78]
	95	Nd	*Aedes aegypti*	[91]
	26.8	>50.0	*Aedes aegypti*	[100]
	27.2 (1 day old larvae)	52.4 (1 day old larvae)	*Aedes aegypti*	[108]
	25 < LC_50_ < 50	50 < LC_90_ < 100	*Aedes aegypti*	[93]
	27.53	Nd	*Aedes aegypti*	[101]
	26	48	*Culex quinquefasciatus*	[84]
	13.44	23.52	*Culex quinquefasciatus*	[98]
	24.70	Nd	*Culex pipiens pallens*	[101]
	29.8	47.5	*Aedes albopictus*	[96]
	22.8	57.4	*Aedes albopictus*	[107]
	22.8	>50.0	*Aedes albopictus*	[100]
	30.03	Nd	*Aedes albopictus*	[101]
	25 < LC_50_ < 50	50 < LC_90_ < 100	*Aedes albopictus*	[81]
	20.21	32.31	*Aedes albopictus*	[112]
	36.42	Nd	*Anopheles sinensis*	[101]
	20.2	32.3	*Aedes albopictus*	[97]
α-Terpinene	14.7	39.3	*Aedes aegypti*	[96]
	28.1	76.4	*Aedes aegypti*	[107]
	0.4	Nd	*Aedes aegypti*	[92]
	12.5 < LC_50_ < 25	12.5 < LC_90_ < 25	*Aedes aegypti*	[93]
	21.30	Nd	*Aedes aegypti*	[101]
	25.2	>50.0	*Aedes albopictus*	[96]
	22.4	58.8	*Aedes albopictus*	[107]
	25 < LC_50_ < 50	<100	*Aedes albopictus*	[81]
	>250	>250	*Culex quinquefasciatus*	[84]
α-Terpineol	>50.0	>50.0	*Aedes aegypti*	[96]
	76.68	Nd	*Aedes aegypti*	[92]
	>100	Nd	*Aedes aegypti*	[93]
	23.49	Nd	*Aedes aegypti*	[101]
	>50.0	>50.0	*Aedes albopictus*	[96]
	21.26	Nd	*Aedes albopictus*	[101]
	>250	>250	*Culex quinquefasciatus*	[84]
	>100	>100	*Culex pipiens pallens*	[88]
	21.30	Nd	*Culex pipiens pallens*	[101]
	194	216	*Culex pipiens molestus*	[94]
	27.16	Nd	*Anopheles sinensis*	[101]
	>500	Nd	*Aedes aegypti*	[79]

Nd: not determined.

**Table 6 molecules-28-02361-t006:** Molluscicidal activity of *Croton hirtus* essential oil and its major components against *Physella acuta* adults (μg/mL).

Material	LC_50_ (95% Limits)	LC_90_ (95% Limits)	χ^2^	*p*
Essential oil	10.09 (8.37–12.21)	17.12 (13.80–25.81)	0.68	0.877
Caryophyllene oxide	5.78 (4.86–6.92)	8.96 (7.38–13.42)	0.50	0.921
α-Humulene	7.24 (6.00–8.67)	11.88 (9.71–17.50)	0.62	0.887
β-Caryophyllene	9.58 (7.79–11.72)	18.08 (14.32–27.14)	0.88	0.829

**Table 7 molecules-28-02361-t007:** Summary of antiparasitic activities of *Croton* spp. essential oils.

Species	Yield (%)	Main Components ^a^	M/S/P	IC_50_ (μg/mL)	SI ^b^	Organisms	Ref.
*Croton argyrophylloides* Müll. Arg.	0.2 to 3	Spathulenol, caryophyllene oxide, β-elemene	M: 0 S: 95.17	15.50 16.71 16.41	>6.45 >6.0 >6.1	Promastigotes of *Leishmania* (V.) *braziliensis* Promastigotes of *Leishmania* (L.) *amazonensis* Promastigotes of *Leishmania* (L.) *chagasi*	[119]
*Croton cajucara* Benth.	Nd	Linalool	Nd	0.0083	Nd	Promastigotes of *Leishmania amazonensis*	[57]
	Nd	Linalool	Nd	0.022	Nd	Amastigotes of *Leishmania amazonensis*	[57]
		Linalool		Essential oil at 15.0 ng/mL was able to kill 100% of the parasites in 60 min.	Nd	Adults of *Leishmania amazonensis.*	[57]
*Croton cajucara* Benth. (white morphotype)	Nd	Linalool, β-caryophyllene, Germacrene D	Nd	1490	Nd	Adults of *Neoechinorhynchus buttnerae*	[120]
*Croton cajucara* Benth. (red morphotype)	Nd	Germacrene D, germacrene A, β-elemene	Nd	1030	Nd	Adults of *Neoechinorhynchus buttnerae*	[120]
*Croton jacobinensis* Müll. Arg.	0.2 to 3	Caryophyllene oxide, spathulenol, germacrene B	M: 0 S: 91.64	23.79 22.06 17.69	>4.2 >4.53 >5.65	Promastigotes of *Leishmania* (V.) *braziliensis* Promastigotes of *Leishmania* (L.) *amazonensis* Promastigotes of *Leishmania* (L.) *chagasi*	[119]
*Croton linearis* Jacq.	1.6	Guaiol	M: 4.89 S: 90.06	20.0	4	Promastigotes of *Leishmania amazonensis*	[121]
				13.8	6.46	Amastigotes of *Leishmania amazonensis*	[121]
				197.26	1.55	Promastigotes of *Trypanosoma cruzi*	[121]
				% Inhibition infection (10 µg/mL): 13.32	Nd	Amastigotes of *Trypanosoma cruzi*	[121]
*Croton linearis* Jacq.	0.9% (*v*/*w*)	1,8-Cineole, α-pinene, sabinene	M: 75.89 S: 24.11	21.4	2	Promastigotes of *Leishmania amazonensis*	[122]
				18.9	3	Amastigotes of *Leishmania amazonensis*	[122]
*Croton macrostachyus* Hochst. ex Delile	0.038	Benzyl benzoate, linalool, γ-muurolene	Ar: 52.5 M: 11.6 S: 34.9	MIC = 0.08 µL/mL	Nd	Promastigotes of *Leishmania donovani*	[123]
				20.00 nL/mL	0.5	Amastigotes of *Leishmania donovani*	[123]
				MIC = 0.16 µL/mL	Nd	Promastigotes of *Leishmania aethiopica*	[123]
				6.66 nL/mL	1.5	Amastigotes of *Leishmania aethiopica*	[123]
*Croton nepetifolius* Baill.	0.2 to 3	Methyl eugenol, β-caryophyllene, 1,8-cineole, germacrene B, 3,5-dimethoxytoluene	M: 14.02 S: 29.18 P: 39.63	9.87 9.08 14.80	>10.13>11.01>6.76	Promastigotes of *Leishmania* (V.) *braziliensis* Promastigotes of *Leishmania* (L.) *amazonensis* Promastigotes of *Leishmania* (L.) *chagasi*	[119]
*Croton piauhiensis* Mull. Arg.	0.04	β-Caryophyllene, caryophyllene oxide, limonene, τ-muurolol, *p*-cymene, bicyclogermacrene	M:39.57 S: 58.85	1.70	Nd	Promastigotes of *Leishmania infantum*	[124]
				13.79	Nd	Axenic amastigotes of *Leishmania infantum*	[124]
*Croton* *pulegiodorus* Baill.	0.27	Ascaridole, *p*-cymene, camphor, isoascaridole	M: 92.9 S: 0	0.05	Nd	Promastigotes of *Leishmania infantum*	[124]
				2.33	Nd	Axenic amastigotes of *Leishmania infantum*	[124]
*Croton rudolphianus* Müll. Arg.	0.96	β-Caryophyllene, bicyclogermacrene, δ-cadinene, germacrene D	M: 8.98 S: 50.94	14.81	Nd	*Schistosoma mansoni* cercariae	[115]
		7-hydroxycalamenene	S	IC_50_: 66.7. MIC: 250	>7.5	Promastigote forms *Leishmania chagasi*	[125]
*Croton sincorensis* Mart. ex Müll. Arg.	0.2 to 3	Caryophyllene oxide, β-eudesmol, spathulenol, hedycaryol, globulol, humulene epoxide II, viridiflorol, 1,8-cineole	M: 8.24 S: 77.92	27.03 14.16 13.05	>3.7 >7.06 >7.66	Promastigotes of *Leishmania* (V.) *braziliensis* Promastigotes of *Leishmania* (L.) *amazonensis* Promastigotes of *Leishmania* (L.) *chagasi*	[119]
*Croton zehntneri* Pax and K. Hoffm.	Nd	(*E*)-Anethole, anisaldehyde, estragole, anisyl acetate	Nd	550 (Ovicidal)	Nd	*Haemonchus contortus*	[28]
	Nd	(*E*)-Anethole, anisaldehyde, estragole, anisyl acetate	Nd	1170 (Larvicidal)	Nd	*Haemonchus contortus*	[28]
*Croton zehntneri* Pax and K. Hoffm.	Nd	(*E*)-Anethole, estragole, germacrene B		740 (Ovicidal)	Nd	*Haemonchus contortus*	[28]
	Nd			1370 (Larvicidal)	Nd	*Haemonchus contortus*	[28]

^a^: The order of the compounds is sorted by percentage from high to low and greater than 5.0%. ^b^: SI: LC_50_ cytotoxicity/LC_50_ parasitic toxicity. Ar: aromatic, M: monoterpenoids, S: sesquiterpenoids, P: phenylpropanoids. Nd = not determined.

**Table 8 molecules-28-02361-t008:** Summary of antiparasitic activities of essential oil components.

Compound	IC_50_/EC_50_/LC_50_ (μg/mL)	Parasites	SI ^a^	Ref.
*(E)*-Anethole	690	Eggs of *Haemonchus contortus*	Nd	[28]
	2110	Larvae of *Haemonchus contortus*	Nd	[28]
Ascaridole	0.1 ± 0.01	Promastigotes of *Leishmania amazonensis*	4	[126]
	0.3 ± 0.05	Amastigotes of *Leishmania amazonensis*	11	[126]
	0.1 ± 0.01	Promastigotes of *Leishmania amazonensis*	4	[127]
	Combination 20:80 mg/kg of ascaridole—carvacrol showed lower (*p* < 0.05) lesion size and parasite burden compared with control groups in in vivo testing on BALB/c mice.	*Leishmania amazonensis*	Nd	[127]
α-Asarone	20.19	Bloodstream forms of *Trypanosoma brucei brucei*	5.21	[128]
Camphor	>100	Bloodstream forms of *Trypanosoma brucei brucei*	Nd	[129]
	IC_50_ > 100	Promastigotes of *Phytomonas davidi*	Nd	[130]
	37.39	Bloodstream forms of *Trypanosoma brucei brucei*	>6.69	[128]
	5.55	Promastigotes of *Leishmania infantum*	4.56	[131]
	7.90	Promastigotes of *Leishmania major*	3.20	[131]
β-Caryophyllene	12.8	Erythrocytic stages *Plasmodium falciparum*	4.86	[132]
	28.9	Bloodstream forms of *Trypanosoma brucei rhodesiense*	2.15	[132]
	50.1	Trypomastigote forms (mammalian stage) of *Trypanosoma cruzi*	1.24	[132]
	52.4	Amastigotes (the clinically relevant form) of *Leishmania donovani*	1.19	[132]
	96 µM	Promastigotes of *Leishmania amazonensis*	Nd	[133]
	13.78	Bloodstream forms of *Trypanosoma brucei brucei*	1.40	[128]
	1.06	Promastigotes of *Leishmania infantum*	20.82	[131]
	1.33	Promastigotes of *Leishmania major*	16.59	[131]
	2.89	Epimastigotes of *Trypanosoma cruzi*	12.93	[134]
	24.54	Intracellular amastigotes infecting Vero cells of *Trypanosoma cruzi*	Nd	[134]
	24.02	Promastigotes of *Leishmania (Leishmania) infantum*	143.85	[134]
	53.39	Intracellular amastigotes infecting THP-1 cells of *Leishmania (Leishmania) infantum*	Nd	[134]
Caryophyllene oxide	4.9	Promastigotes of *Leishmania amazonensis*	0.92	[126]
	4.4	Amastigotes of *Leishmania amazonensis*	1.0	[126]
	IC_50_ > 100	Promastigotes of *Phytomonas davidi*	Nd	[130]
	17.70	Bloodstream forms of *Trypanosoma brucei brucei*	2.14	[128]
	4.9	Promastigotes of *Leishmania amazonensis*	Nd	[127]
1,8-Cineole	At 200 μg/mL it killed 100% of protoscoleces after 30 min.	Protoscoleces of *Echinococcus granulosus*	Nd	[135]
	568.1	Promastigotes of *Leishmania amazonensis*	>0.18	[136]
	>100	Bloodstream forms of *Trypanosoma brucei brucei*	Nd	[129]
	IC_50_ > 100	Promastigotes of *Phytomonas davidi*	Nd	[130]
	83.15	Bloodstream forms of *Trypanosoma brucei brucei*	>3.00	[128]
	Inactive	Promastigotes of *Leishmania infantum*	Nd	[137]
	Inactive	Promastigotes of *Leishmania tropica*	Nd	[137]
	Inactive	Promastigotes of *Leishmania major*	Nd	[137]
	53.40	Promastigotes of *Leishmania infantum*	5.74	[131]
	74.80	Promastigotes of *Leishmania major*	4.10	[131]
	0.63	Epimastigotes of *Trypanosoma cruzi*	63.49	[134]
	>100	Intracellular amastigotes infecting Vero cells of *Trypanosoma cruzi*	Nd	[134]
	>100	Promastigotes of *Leishmania (Leishmania) infantum*	Nd	[134]
	>100	Intracellular amastigotes infecting THP-1 cells of *Leishmania (Leishmania) infantum*	Nd	[134]
*p*-Cymene	>20	Erythrocytic stages *Plasmodium falciparum*	4.5	[132]
	45.0	Bloodstream forms of *Trypanosoma brucei rhodesiense*	2.0	[132]
	>90	Trypomastigote forms (mammalian stage) of *Trypanosoma cruzi*	Nd	[132]
	>90	Amastigotes (the clinically relevant form) of *Leishmania donovani*	Nd	[132]
	>1000	Promastigotes of *Leishmania amazonensis*	Nd	[136]
	76.32	Bloodstream of *Trypanosoma brucei brucei*	>0.66	[138]
	IC_50_ > 100	Promastigotes of *Phytomonas davidi*	Nd	[130]
	156.17	Promastigotes of *Leishmania infantum*	Nd	[131]
	219.17	Promastigotes of *Leishmania major*	Nd	[131]
(−)-α-Bisabolol	20μM	Promastigotes of *Trypanosoma cruzi*	26.5	[139]
	285μM	Epimastigote of *Trypanosoma cruzi*	2.05	[139]
	Topical treatment at 2.5% reduced lesion thickness to 56% and had a higher efficacy than the reference control, meglumine antimoniate.	*Leishmania tropica*	Nd	[117]
	5.9	Amastigotes of *Leishmania amazonensis*	5.41	[140]
	4.8	Amastigotes of *Leishmania infantum*	6.65	[140]
Borneol	>20	Erythrocytic stages *Plasmodium falciparum*	4.5	[132]
	24.3	Bloodstream forms of *Trypanosoma brucei rhodesiense*	3.70	[132]
	>90	Trypomastigote forms (mammalian stage) of *Trypanosoma cruzi*	Nd	[132]
	52.1	Amastigotes (the clinically relevant form) of *Leishmania donovani*	1.73	[132]
	>100	Bloodstream forms of *Trypanosoma brucei brucei*	Nd	[129]
	Inactive	Promastigotes of *Leishmania infantum*	Nd	[137]
	Inactive	Promastigotes of *Leishmania tropica*	Nd	[137]
	Inactive	Promastigotes of *Leishmania major*	Nd	[137]
Eugenol	82.9	Promastigotes of *Leishmania amazonensis*	Nd	[136]
	>100	Bloodstream forms of *Trypanosoma brucei brucei*	Nd	[129]
	60.4	Amastigotes of *Leishmania braziliensis*	1.3	[141]
	43.8	Amastigotes of *Trypanosoma cruzi*	1.8	[141]
	665.6	Amastigotes of *Plasmodium falciparum*	0.12	[141]
	37.20	Bloodstream forms of *Trypanosoma brucei brucei*	2.50	[128]
	80	Promastigote forms of *Leishmania amazonensis*	Nd	[142]
Estragole	32.08	Bloodstream forms of *Trypanosoma brucei brucei*	>7.80	[128]
α-Humulene	9.76	*Leishmania donovani*	Nd	[143]
*R*-(+)-Limonene	4.24	Bloodstream of *Trypanosoma brucei brucei*	>11.79	[138]
	35.55	Bloodstream forms of *Trypanosoma brucei brucei*	4.50	[128]
	14.1	Trypomastigote forms of *Trypanosoma cruzi*	Nd	[144]
	33.7	Epimastigotes of *Trypanosoma cruzi*	Nd	[144]
Limonene	At a concentration of 43.75 µg/mL it produced decreased motility.	Adult worms of *Schistosoma mansoni*	Nd	[145]
	278 µM	Promastigotes of *Leishmania amazonensis*	Nd	[133]
	252.0 μM.	Promastigotes of *Leishmania amazonensis*		
	147.0 μM	Amastigote of *Leishmania amazonensis*		
	354.0 μM	Promastigotes of *Leishmania major*		
	185.0 μM	Promastigotes of *Leishmania braziliensis*		
	201.0 μM	Promastigotes of *chagasi*		
	38.71	Epimastigotes of *Trypanosoma cruzi*	>100	[134]
	145.94	Intracellular amastigotes infecting Vero cells of *Trypanosoma cruzi*	Nd	[134]
	>100	Promastigotes of *Leishmania (Leishmania) infantum*	Nd	[134]
	>100	Intracellular amastigotes infecting THP-1 cells of *Leishmania (Leishmania) infantum*	Nd	[134]
(−)-Linalool	>20	Erythrocytic stages *Plasmodium falciparum*	4.5	[132]
	3.6	Bloodstream forms of *Trypanosoma brucei rhodesiense*	25	[132]
	>90	Trypomastigote forms (mammalian stage) of *Trypanosoma cruzi*	Nd	[132]
	86.3	Amastigotes (the clinically relevant form) of *Leishmania donovani*	1.04	[132]
	276.2	Promastigotes of *Leishmania amazonensis*	Nd	[136]
(±)-Linalool	39.32	Bloodstream forms of *Trypanosoma brucei brucei*	5.20	[128]
Linalool	430	Promastigotes of *Leishmania braziliensis*	16.93	[146]
	Nd	Amastigote of *Leishmania braziliensis*	Nd	[146]
	>100	Bloodstream forms of *Trypanosoma brucei brucei*	Nd	[129]
	198.6	Epimastigote of *Trypanosoma cruzi*	>5	[147]
	249.6	Intracellular amastigote of *Trypanosoma cruzi*	>4	[147]
	IC_50_ > 100	Promastigotes of *Phytomonas davidi*	Nd	[130]
	30.16	Epimastigotes of *Trypanosoma cruzi*	26.33	[134]
	>100	Intracellular amastigotes infecting Vero cells of *Trypanosoma cruzi*	Nd	[134]
	>100	Promastigotes of *Leishmania (Leishmania) infantum*	Nd	[134]
	>100	Intracellular amastigotes infecting THP-1 cells of *Leishmania (Leishmania) infantum*	Nd	[134]
	0.31	Trypomastigote forms of *Trypanosoma cruzi*	2.7	[148]
	0.0043	Promastigotes of *Leishmania amazonensis*	Nd	[57]
	0.0155	Amastigote of *Leishmania amazonensis*	Nd	[57]
Myrcene	>20	Erythrocytic stages *Plasmodium falciparum*	4.5	[132]
	22	Bloodstream forms of *Trypanosoma brucei rhodesiense*	4.10	[132]
	>90	Trypomastigote forms (mammalian stage) of *Trypanosoma cruzi*	Nd	[132]
	48.2	Amastigotes (the clinically relevant form) of *Leishmania donovani*	1.87	[132]
	2.24	Bloodstream of *Trypanosoma brucei brucei*	>22.32	[138]
Nerolidol	74.3	Promastigotes of *Leishmania braziliensis*	20.19	[146]
	47.5	Amastigote of *Leishmania braziliensis*	2.20	[146]
	85	Promastigotes of *Leishmania amazonensis*	Nd	[118]
	67	Amastigote of *Leishmania amazonensis*	Nd	[118]
	74	Promastigotes of *Leishmania braziliensis*	Nd	[118]
	75	Promastigotes of *Leishmania chagasi*	Nd	[118]
	*Leishmania-amazonensis*-infected BALB/c mice were treated with intraperitoneal doses of 100 mg/kg/day for 12 days or topically with 5 or 10% ointments for 4 weeks, and both resulted in significant reductions in lesion sizes.	*Leishmania amazonensis*	Nd	[118]
(*Z*)-Nerolidol	15.78	Bloodstream forms of *Trypanosoma brucei brucei*	1.87	[128]
α-Phellandrene	9.2	Bloodstream of *Trypanosoma brucei*	2.9	[149]
	32.8	Promastigotes of *Leishmania major*	0.8	[149]
α-Pinene	10.7	Erythrocytic stages *Plasmodium falciparum*	8.21	[132]
	0.42	Bloodstream forms of *Trypanosoma brucei rhodesiense*	209.05	[132]
	>90	Trypomastigote forms (mammalian stage) of *Trypanosoma cruzi*	Nd	[132]
	81.9	Amastigotes (the clinically relevant form) of *Leishmania donovani*	1.07	[132]
	4.1	Bloodstream form of *Trypanosoma brucei*	0.6	[149]
	55.3	Promastigotes of *Leishmania major*	<0.1	[149]
	2.9	Bloodstream forms of *Trypanosoma brucei brucei*	>34.5	[129]
	1.145	Tachyzoites of *Toxoplasma gondii* RH strain	126	[150]
	17.60	Promastigotes of *Leishmania infantum*	13.08	[131]
	19.80	Promastigotes of *Leishmania major*	11.63	[131]
	2.74	Epimastigotes of *Trypanosoma cruzi*	11.57	[134]
	1.92	Intracellular amastigotes infecting Vero cells of *Trypanosoma cruzi*	Nd	[134]
	45.94	Promastigotes of *Leishmania (Leishmania) infantum*	57.25	[134]
	>100	Intracellular amastigotes infecting THP-1 cells of *Leishmania (Leishmania) infantum*	Nd	[134]
β-Pinene	47.37	Bloodstream form of *Trypanosoma brucei brucei*	>1.06	[138]
	54.8	Bloodstream form of *Trypanosoma brucei*	0.5	[149]
	200.1	Promastigotes of *Leishmania major*	0.1	[149]
	0.326	Tachyzoites of *Toxoplasma gondii* RH strain	61	[150]
	50 < IC_50_ < 100	Promastigotes of *Phytomonas davidi*	Nd	[130]
Sabinene	17.7	Bloodstream of *Trypanosoma brucei*	1.3	[149]
	126.6	Promastigotes of *Leishmania major*	0.2	[149]
α-Terpinene	3.7	Erythrocytic stages *Plasmodium falciparum*	22.89	[132]
	3.1	Bloodstream forms of *Trypanosoma brucei rhodesiense*	27.32	[132]
	49.1	Trypomastigote forms (mammalian stage) of *Trypanosoma cruzi*	1.73	[132]
	10.5	Amastigotes (the clinically relevant form) of *Leishmania donovani*	8.07	[132]
γ-Terpinene	IC_50_ > 100	Promastigotes of *Phytomonas davidi*	Nd	[130]
	>20	Erythrocytic stages *Plasmodium falciparum*	4.5	[132]
	32.9	Bloodstream forms of *Trypanosoma brucei rhodesiense*	2.74	[132]
	>90	Trypomastigote forms (mammalian stage) of *Trypanosoma cruzi*	Nd	[132]
	>90	Amastigotes (the clinically relevant form) of *Leishmania donovani*	Nd	[132]
Terpinen-4-ol	>20	Erythrocytic stages *Plasmodium falciparum*	2.17	[132]
	0.66	Bloodstream forms of *Trypanosoma brucei rhodesiense*	65.61	[132]
	46.8	Trypomastigote forms (mammalian stage) of *Trypanosoma cruzi*	0.93	[132]
	68.7	Amastigotes (the clinically relevant form) of *Leishmania donovani*	0.66	[132]
	0.02	Bloodstream form of *Trypanosoma brucei*	1025.0	[149]
	335.9	Promastigotes of *Leishmania major*	0.1	[149]
(−)-Terpinen-4-ol	39.51	Bloodstream forms of *Trypanosoma brucei brucei*	2.64	[128]
α-Terpineol	>20	Erythrocytic stages *Plasmodium falciparum*	1.62	[132]
	0.56	Bloodstream forms of *Trypanosoma brucei rhodesiense*	57.68	[132]
	61.0	Trypomastigote forms (mammalian stage) of *Trypanosoma cruzi*	0.53	[132]
	75.9	Amastigotes (the clinically relevant form) of *Leishmania donovani*	0.43	[132]

^a^: SI: LC_50_ cytotoxicity/LC_50_ parasitic. Nd: not determined.

**Table 9 molecules-28-02361-t009:** Antimicrobial activity of *Croton hirtus* essential oil and major compounds (MIC, IC_50_, and µg/mL).

Material	Gram-Positive			Gram-Negative			Yeast
	*E. faecalis* ATCC29212	*S. aureus* ATCC25923	*B. cereus* ATCC13245	*E. coli* ATCC25922	*P. aeruginosa* ATCC27853	*S. enterica* ATCC13076	*C. albicans* ATCC10231
			MIC (µg/mL)				
Essential oil	8	16	16	16	Na	16	Na
β-Caryophyllene	32	64	64	64	Na	64	Na
α-Humulene	8	16	32	32	Na	16	Na
Caryophyllene oxide	8	32	32	32	Na	32	Na
Streptomycin	256	256	128	32	256	128	Nt
Kanamycin	128	4	8	128	64	16	Nt
Tetracycline	4	16	64	8	256	64	Nt
Nystatin	Nt	Nt	Nt	Nt	Nt	Nt	4
Cyclohexamide	Nt	Nt	Nt	Nt	Nt	Nt	32
			IC_50_ (µg/mL)				
Essential oil	3.12 ± 1.36	5.34 ± 0.98	5.23 ± 0.21	5.67 ± 1.45	Na	5.98 ± 0.09	Na
β-Caryophyllene	9.35 ± 2.34	21.23 ± 1.35	18.56 ± 1.32	21.46 ± 1.34	Na	20.15 ± 1.48	Na
α-Humulene	3.24 ± 2.12	5.34 ± 1.34	9.35 ± 0.36	10.45 ± 1.56	Na	5.23 ± 0.08	Na
Caryophyllene oxide	2.67 ± 2.00	9.56 ± 1.43	9.32 ± 0.21	12.56 ± 2.56	Na	9.34 ± 0.91	Na
Streptomycin	50.34 ± 2.32	45.24 ± 1.36	20.45 ± 0.39	9.45 ± 0,35	68.67 ± 1.89	45.67 ± 2.30	Nt
Cyclohexamide	Nt	Nt	Nt	Nt	Nt	Nt	10.46 ± 0.32

Na: Not active; Nt: not tested.

**Table 10 molecules-28-02361-t010:** Summary of antimicrobial activity of *Croton* spp. essential oils.

Species	Yield (%)	Main Components ^a^	M/S/P/B or Other (%)	MIC or IC_50_ (μg/mL)	Organisms	Ref.
*Croton* *adamantinus* Müll. Arg.	0.6	Methyl eugenol, 1,8-cineole, bicyclogermacrene, β-caryophyllene.	M: 27.66 S: 32.42 P: 14.81	Synergistic effect with gentamicin.	*Enterobacter aerogenes, Pseudomonas aeruginosa,* Methicillin-resistant *Staphylococcus aureus.*	[154]
				Synergistic effect with amoxicillin + clavulanate.	Methicillin-resistant *Staphylococcus aureus.*	[154]
				Synergistic effect with cefepime.	*Enterobacter aerogenes,* Methicillin-resistant *Staphylococcus aureus.*	[154]
*Croton adipatus* Kunth	0.47 ± 0.01	β-Myrcene; α-thujene; limonene; α-phellandrene, β-elemene.	M: 72.73 S: 18.82	>1000	*Staphylococcus aureus*	[155]
				286.4	*Bacillus subtilis*	
				>1000	*Escherichia coli*	
				>1000	*Pseudomonas aeruginosa*	
				572.8	*Candida albicans*	
*Croton argyrophyllus* Kunth	Nd	Bicyclogermacrene, β-pinene, spathulenol, β-caryophyllene, β-phellandrene.	M: 27.94 S: 62.14	10	*Bacillus cereus*	[156]
				25	*Bacillus subtilis*	
				25	*Staphylococcus aureus*	
				25	*Escherichia coli*	
				25	*Pseudomonas aeruginosa*	
				Nd	*Candida albicans*	
				Nd	*Candida glabrata*	
				Nd	*Candida parapsilosis*	
*Croton argyrophyllus* Kunth	0.1 to 0.7	Bicyclogermacrene, *epi*-longipinanol, spathulenol.	M: 0 S: 99.17–100	312	*Staphylococcus aureus*	[157]
				NI	*Escherichia coli*	
	0.1 to 0.7	Bicyclogermacrene, (*Z*)-caryophyllene, *epi*-longipinanol, germacrene B, guaiol, 10-*epi*-γ-eudesmol, α-muurolol.	M: 0 S: 99.1–99.61	78	*Staphylococcus aureus*	[157]
				≥1024	*Escherichia coli*	
	0.1 to 0.7	Bicyclogermacrene, (*Z*)-caryophyllene, germacrene B, *epi*-longipinanol.	M: 0 S: 100	156	*Staphylococcus aureus*	[157]
				NI	*Escherichia coli*	
*Croton argyrophyllus* Kunth	0.38	α-Pinene, bicyclogermacrene	M: 68.5 S: 29.87	Synergistic effect with chlorhexidine.	*Streptococcus mutans* *Streptococcus salivarius* *Streptococcus sanguinis*	[158]
*Croton argyrophylloides* Müll. Arg. (syn *Croton tricolor* Baill.)	Nd	Spathulenol, bicyclogermacrene, 1,8-cineole, β-elemene, β-caryophyllene, α-pinene.		NI	*Candida albicans*	[56]
				NI	*Candida tropicalis*	
				9–19	*Microsporum canis*	
*Croton argyrophylloides* Müll. Arg	0.5	Bicyclogermacrene, spathulenol, β-caryophyllene, myrcene, α-pinene, β-phellandrene, 1,8-cineole.	M: 48.22 S: 47.7	97–195	Forty-nine clinical strains of *Mycobacteria tuberculosis*	[159]
				97	Standard strain H_37_RV of *Mycobacteria tuberculosis*	[159]
*Croton blanchetianus* Baill.	Nd	Caryophyllene oxide, δ-amorphene, τ-muurolol, 1,8-cineole.	M: 10.05 S: 38.84	Inhibition of planktonic cells growth at 50 µg/mL: 78%.	*Candida albicans*	[160]
				Inhibition of planktonic cells growth at 50 µg/mL: 75%.	*Candida parapsilosis*	
*Croton blanchetianus* Baill.	7.5	α-Pinene, eucalyptol, sativene, β-caryophyllene, bicyclogermacrene, spathulenol.	M: 35.55–38.07 S: 55.45–55.95	Inactivated at a concentration of 900 µg/mL	*Listeria monocytogenesStaphylococcus aureus* *Leuconostoc mesenteroides* *Weissella viridescens*	[161]
*Croton cajucara* Benth.	0.4	Linalool	Nd	22.3	*Lactobacillus casei*	[162]
				13.8	*Streptococcus sobrinus*	
				40.1	*Streptococcus mutans*	
				31.2	*Porphyromonas gingivalis*	
				33.4	*Staphylococcus aureus*	
				13.4	*Candida albicans*	
*Croton cajucara* Benth.	Nd	7-Hydroxycalamenene, δ-cadinene, γ-cadinene, germacrene B, τ-cadinol, caryophyllene oxide.	M: 0 S: 97.59	12.21	*Absidia cylindospora*	[163]
*Croton cajucara* Benth.	0.8	Linalool, 7-hydroxycalamenene, β-caryophyllene, germacrene D.	Dominated by Sesquiterpenes	Inhibition growth zones (in cm): 0.9–1.3.	*Candida albicans*	[164]
			Dominated by Sesquiterpenes	Inhibition growth zones (in cm): 0.5–1.6.	*Staphylococcus aureus*	[164]
	1.0	Linalool, nerolidol, β-caryophyllene, bicyclogermacrene germacrene D.	Dominated by Sesquiterpenes	Inhibition growth zones (in cm): 0.9–1.3	*Candida albicans*	[164]
		Linalool, nerolidol, β-caryophyllene, bicyclogermacrene germacrene D.	Dominated by Sesquiterpenes	Inhibition growth zones (in cm): 0.2–1.0	*Staphylococcus aureus*	[164]
*Croton cajucara* Benth.	0.65	7-Hydroxycalamenene, α-pinene, linalool.	Nd	39.06	*Mycobacterium smegmatis*	[165]
				4.88	*Mycobacterium tuberculosis*	
				0.019	Methicillin-resistant *Staphylococcus aureus*	
				1.22	*Candida albicans*	
				Nd	*Mucor circinelloides*	
				Nd	*Rhizopus oryzae*	
		α-Pinene, linalool, β-caryophyllene.	Nd	5000	*Mycobacterium smegmatis*	[165]
				4.88	*Mycobacterium tuberculosis*	
				Na	Methicillin-resistant *Staphylococcus aureus*	
				1250	*Candida albicans*	
				Nd	*Mucor circinelloides*	
				Nd	*Rhizopus oryzae*	
		7-Hydroxycalamenene (28.4%), linalool (11.0%).	Nd	78.12	*Mycobacterium smegmatis*	[165]
				4.88	*Mycobacterium tuberculosis*	
				0.019	Methicillin-resistant *Staphylococcus aureus*	
				156.25	*Candida albicans*	
				Nd	*Mucor circinelloides*	
				Nd	*Rhizopus oryzae*	
		7-Hydroxycalamenene (30.9%), linalool (9.9%).	Nd	156.25	*Mycobacterium smegmatis*	[165]
				4.88	*Mycobacterium tuberculosis*	
				0.004	Methicillin-resistant *Staphylococcus aureus*	
				0.001	*Candida albicans*	
				Nd	*Mucor circinelloides*	
				Nd	*Rhizopus oryzae*	
		7-Hydroxycalamenene (32.9%), linalool (13.2%).	Nd	156.25	*Mycobacterium smegmatis*	[165]
				4.88	*Mycobacterium tuberculosis*	
				0.001	Methicillin-resistant *Staphylococcus aureus*	
				0.38	*Candida albicans*	
				3.63 × 10^−8^	*Mucor circinelloides*	
				0.152	*Rhizopus oryzae*	
		7-Hydroxycalamenene.	S	39.06	*Mycobacterium smegmatis*	[165]
				312.5	*Mycobacterium tuberculosis*	
				39.06	Methicillin-resistant *Staphylococcus aureus*	
				78.125	*Candida albicans*	
				19.53	*Mucor circinelloides*	
				39.06	*Rhizopus oryzae*	
*Croton campestris* A. St.Hil.	0.04 (leaves)	β-Caryophyllene, bicyclogermacrene, limonene, τ-cadinol.	M: 28.1 S: 67.6	≥512	*Escherichia coli*	[9]
				≥512	*Staphylococcus aureus*	
				≥1024	*Shigella flexneri*	
				≥1024	*Pseudomonas aeruginosa*	
				≥1024	*Bacillus cereus*	
	0.02 (branches)	Spathulenol, bicyclogermacrene, β-caryophyllene, terpinen-4-ol, murola-4,10(14)-dien-1-ol.	M: 25.1 S: 67.8	≥512	*Escherichia coli*	[9]
				≥128	*Staphylococcus aureus*	
				≥512	*Shigella flexneri*	
				≥512	*Pseudomonas aeruginosa*	
				≥256	*Bacillus cereus*	
				Synergistic effect with gentamicin	*Staphylococcus aureus, Shigella flexneri.*	
				Synergistic effect with neomycin.	*Pseudomonas aeruginosa, Bacillus cereus*	
				Synergistic effect with kanamycin.	*Staphylococcus aureus, Bacillus cereus*	
*Croton campestris* St. Hilaire.	0.40	Caryophyllene oxide, humulene oxide II.	M: 16.9 S: 75.2	1.56	*Staphylococcus aureus*	[166]
				6.25	*Enterrococcus hirae*	
				6.25	*Candida albicans*	
*Croton ceanothifolius* Baill.	0.23	Bicyclogermacrene, germacrene D, β-caryophyllene, 1,10-di-*epi*-cubebol.	M: 8.7 S: 91.3	Synergistic effect with norfloxacin, gentamicin, penicillin.	*Staphylococcus aureus, Pseudomonas aeruginosa, Escherichia coli.*	[10]
*Croton ciliatoglandulifer* Ortega.	Nd	Caryophyllene oxide, cubenol, β-caryophyllene.	M: 3.5 S: 91.3	500	*Candida albicans*	[167]
*Croton conduplicatus* Kunth.		1,8-Cineole, *p*-cymene, β-caryophyllene, spathulenol.	M: 51.31 S: 44.42	256 512	Methicillin-sensitive *Staphylococcus aureus* Methicillin-resistant *Staphylococcus aureus*	
				Synergistic effect with ampicillin.	Methicillin-sensitive *Staphylococcus aureus* Methicillin-resistant *Staphylococcus aureus*	[168]
*Croton doctoris* S. Moore.	0.4	β-Caryophyllene; caryophyllene oxide; α-humulene; α-selinene.	M: 0 S: 83.32	0.625 (*v*/*v*)	Streptococci group	[169]
*Croton ferrugineus* Kunth.	0.06 ± 0.02	β-Caryophyllene, limonene + β-phellandrene, myrcene, germacrene D, linalool, α-humulene.	M: 47.03 S: 47.63	>2000	*Escherichia coli*	[170]
				>2000	*Enterococcus faecalis*	
				2000	*Micrococcus luteus*	
				>2500	*Staphylococcus aureus*	
				1000	*Cándida albicans*	
*Croton ferrugineus* Kunth.	0.06 ± 0.001	β-Caryophyllene, limonene, β-thujene, β-myrcene, β-elemene.	M: 28.03 S: 70.26	>1000	*Staphylococcus aureus*	[155]
				72	*Bacillus subtilis*	
				>1000	*Escherichia coli*	
				>1000	*Pseudomonas aeruginosa*	
				576.2	*Candida albicans*	
*Croton gratissimus* Burch.		Sabinene, α-phellandrene, β-phellandrene, α-pinene, germacrene D.	M: 64.8 S: 27.3	1300	*Bacillus cereus*	[171]
				600	*Staphylococcus aureus*	
				200	*Staphylococcus faecalis*	
				1300	*Escherichia coli*	
				2500	*Proteus vulgaris*	
				5000	*Pseudomonas aeruginosa*	
				5000	*Kiebsiella pneumoniae*	
				>10,000	*Proteus vulgaris*	
				>10,000	*Enterobacter cloacae*	
*Croton grewioides* Baill.	0.1	α-Pinene, sabinene, limonene, bicyclogermacrene, β-caryophyllene.	M: 55.56 S: 44.44	Synergistic effect with norfloxacin, tetracycline.	*Staphylococcus aureus*	[172]
*Croton heliotropiifolius* Kunth.	Nd	Limonene, α-pinene, β-caryophyllene, bicyclogermacrene, γ-terpinene.	M: 62.23 S: 35.27	NI	*Bacillus cereus*	[156]
				NI	*Bacillus subtilis*	
				NI	*Staphylococcus aureus*	
				NI	*Escherichia coli*	
				NI	*Pseudomonas aeruginosa*	
				NI	*Candida albicans*	
				NI	*Candida glabrata*	
				NI	*Candida parapsilosis*	
*Croton heliotropiifolius* Kunth (Summer, February).	0.36	β-Caryophyllene, bicyclogermacrene, 1,8-cineole, limonene.	M: 31.72 S: 64.86	500	*Bacillus cereus*	[173]
				6.25	*Enterococcus faecalis*	
				500	*Escherichia coli*	
				Nd	*Klebsiella pneumoniae*	
				500	*Salmonella enterica*	
				500	*Serratia marcescens*	
				500	*Shigella flexneri*	
				Nd	*Staphylococcus aureus*	
*Croton heliotropiifolius* Kunth (Autumn, May).	0.16	β-Caryophyllene, 1,8-cineole, limonene, bicyclogermacrene.	M: 41.12 S: 50.96	Nd	*Bacillus cereus*	[173]
				125	*Enterococcus faecalis*	
				Nd	*Escherichia coli*	
				Nd	*Klebsiella pneumoniae*	
				Nd	*Salmonella enterica*	
				500	*Serratia marcescens*	
				500	*Shigella flexneri*	
				Nd	*Staphylococcus aureus*	
*Croton heliotropiifolius* Kunth (Winter, August).	0.60	β-Caryophyllene, bicyclogermacrene, germacrene D.	M: 16.05 S: 82.39	Nd	*Bacillus cereus*	[173]
				500	*Enterococcus faecalis*	
				500	*Escherichia coli*	
				Nd	*Klebsiella pneumoniae*	
				Nd	*Salmonella enterica*	
				500	*Serratia marcescens*	
				Nd	*Shigella flexneri*	
				Nd	*Staphylococcus aureus*	
*Croton heliotropiifolius* Kunth (Spring, November).	0.24	β-Caryophyllene, bicyclogermacrene, germacrene D.	M: 6.04 S: 84.74	Nd	*Bacillus cereus*	[173]
				500	*Enterococcus faecalis*	
				500	*Escherichia coli*	
				Nd	*Klebsiella pneumoniae*	
				500	*Salmonella enterica*	
				500	*Serratia marcescens*	
				Nd	*Shigella flexneri*	
				Nd	*Staphylococcus aureus*	
*Croton heliotropiifolius* Kunth.		β-Caryophyllene, γ-muurolene, viridiflorene.	M: 2.01 S: 77.14	˃500	*Micrococcus luteus*	[174]
				500	*Sthaphylococcus* *aureus*	[174]
				62.5	*Bacillus subtilis*	[174]
				˃500	*Escherichia coli*	[174]
				˃500	*Pseudomonas aeruginosa*	[174]
				˃500	*Salmonella choleraesuis*	[174]
*Croton heterocalyx* Baill.	0.45	Germacrene D, bicyclogermacrene, δ-elemene, β-elemene, spathulenol, linalool.	M: 13.9 S: 84.8	2800 μg/mL	*Aspergillus niger* *Candida albicans* *Pseudomonas aeruginosa* *Escherichia coli* *Staphylococcus aureus*	[175]
*Croton hieronymi* Griseb.	0.07	γ-Asarone, (*E*)-asarone, borneol, camphor.	M: 35.4 S: 9.9 P: 37.1	Percentage of living microorganism: 0% at 100 μg/mL.	*Escherichia coli* *Candida albicans*	[176]
				Percentage of living microorganism: 50% at 1000 μg/mL.	*Salmonella typhimurium*	
				Percentage of living microorganism: 50% at 100 μg/mL.	*Klebsiella pneumoniae*	
*Croton hirtus* L’ Hér.	0.60	β-caryophyllene, germacrene D, α-humulene, β-elemene.	M: 15.55 S: 77.94	>512	*Escherichia coli*	[8]
				512	*Staphylococcus aureus*	
				Synergistic effect with gentamicin, ceftazidime.	*Staphylococcus aureus*	
*Croton limae* A.P. Gomes.	0.36	Cedrol, 1,8-cineole, α-pinene.	M: 42.4 S: 41	512	*Staphylococcus aureus*	[24]
				≥1024	*Escherichia coli*	
				≥1024	*Pseudomonas aeruginosa*	
				≥1024	*Klebsiella pneumoniae*	
				≥1024	*Candida tropicalis*	
				≥1024	*C. krusei*	
				≥1024	*C. albicans*	
*Croton lechleri* Müll. Arg.	0.061	Sesquicineole, α-calacorene.	M: 18.84 S: 76.82	10,100	*Pseudomonas aeruginosa*	[177]
				1010	*Klebsiella oxytoca*	
				100	*Escherichia coli*	
				10,100	*Staphylococcus aureus* subsp. *aureus*	
				10,100	*Enterococcus foecalis*	
				10,100	*Micrococcus luteus*	
*Croton malambo* H. Karst.	Nd	Methyl eugenol.	M: 0.8 S: 3.3 P: 95.1	Inhibition zones in mm from 7.0–8.0 at 10 mg/mL.	*Staphylococcus aureus* *Candida tropicalis*	[55]
*Croton monteverdensis* Huft.	0.03	α-Pinene, β-pinene, linalool.	M: 47.9 S: 51.0	625 156	*Bacillus cereus* *Staphylococcus aureus*	[178,179]
*Croton niveus* Jacq.	0.10	α-Pinene, 1,8-cineole, borneol.	M: 78.3 S: 19.1	625 78	*Bacillus cereus* *Staphylococcus aureus*	[178,179]
*Croton nepetifolius* Baill.		Methyl eugenol, bicyclogermacrene, β-caryophyllene, *trans*-α-bergamotene, 1,8-cineole, α-humulene, *ortho*-vanillin.		NI	*Candida albicans*	[56]
				NI	*Candida tropicalis*	
				>5000	*Microsporum canis*	
*Croton oblongifolius* Roxb.	0.9	Terpinen-4-ol; α-guaiene; α-bulnesene; β-caryophyllene; myrcene; cyclosativene.	M: 40.3 S: 47.2	0.125%, *v*/*v*	*Propionibacterium acnes*	[180]
*Croton piauhiensis* Müll. Arg.	0.02	β-caryophyllene, limonene, γ-terpinene, germacrene D.	Nd	0.15 (*v*/*v*) 1.25 (*v*/*v*)	*Staphylococcus aureus Staphylococcus aureus* (methicillin-resistant)	[49]
				5.0 (*v*/*v*)	*Staphylococcus epidermidis*	
				>5.0 (*v*/*v*)	*Pseudomonas aeruginosa*	
				5.0 (*v*/*v*)	*Escherichia coli*	
*Croton pluriglandulosus.*	0.46	1,8-Cineole, methyleugenol, elemicin, β-caryophyllene, bicyclogermacrene, 1,3,5-trimethoxybenzene, 3,5-dimethoxytoluene.	M: 6.57 S: 24.83 B: 48.98	Synergistic effect with chlorhexidine.	*Streptococcus mutans* *Streptococcus salivarius* *Streptococcus sanguinis*	[158]
*Croton rhamnifolioides.*	Nd	Spathulenol, 1,8-cineole, *o*-cymene, α-terpineol, *trans*-caryophyllene.	M: 45.65 S: 51.06	1024 Synergistic with antibiotics aminoglycoside and β-lactam, and the antifungal polyene.	*Escherichia coli* *Staphylococcus aureus* *Pseudomonas aeruginosa* *Candida albicans* *Candida krusei* *Candida tropicalis*	[181]
*Croton stellulifer* B.L. Burtt.	0.25–0.44	α-Phellandrene, *p*-cymene, linalool, α-pinene.	M: 73.5–77.4 S: 5.1–5.4	Inhibition zones in mm from 9.3–17.3.	*Escherichia coli* *Staphylococcus aureus* *Staphylococcus faecalis* *Staphylococcus epidermidis* *Proteus vulgaris* *Cryptococcus neoforomans* *Cladosporium cladosporioidesAspergillus fumigatus*	[182]
*Croton tetradenius* Baill.	2.4–4.9	*p*-Cymene, camphor, 1,8-cineole, γ-terpinene, *trans*-ascaridole, *cis*-ascaridole.	M: 94.05 S: 2.52	125	*Staphylococcus aureus*	[183]
				31.5	*Bacillus cereus*	
				250	*Escherichia coli*	
				62.5	*Listeria monocytogenes*	
				125	*Salmonella typhimurium*	
*Croton tetradenius* Baill.	2.4–4.9	Camphor, *p*-cymene, *trans*-ascaridole, *trans*-pinocarveol, 1,8-cineole, α-pinene, pinocarvone.	M: 93.22 S: 1.34	125	*Staphylococcus aureus*	[183]
				31.25	*Bacillus cereus*	
				250	*Escherichia coli*	
				62.5	*Listeria monocytogenes*	
				125	*Salmonella typhimurium*	
*Croton tetradenius* Baill. (CTE101)	4.0	Camphor, *p*-cymene, *trans*-ascaridole.	M: 93.22 S: 1.34	5600	*Escherichia coli*	[184]
				11,300	*Staphylococcus aureus*	
				11,300	*Klebsiella pneumoniae*	
*Croton tetradenius* Baill. (CTE407)	4.0	*p*-Cymene, *trans*-ascaridole, 1,8-cineole, camphor, α-terpinene, γ-terpinene, *cis*-ascaridole.	M: 95.94 S: 1.39	2800	*Escherichia coli*	[184]
				2800	*Staphylococcus aureus*	
				5600	*Klebsiella pneumoniae*	
*Croton tetradenius* Baill.	0.47	*p*-Cymene, camphor, α-phellandrene, γ-terpinene, α-terpinene, *trans*-chrysanthenyl acetate.	M: 99.34 S: 0.66	4000	*Staphylococcus aureus*	[185]
*Croton tetradenius* Baill.	0.27	*trans*-Chrysanthenyl acetate, α-terpinene, *p*-cymene, γ-terpinene.	M: 87.49 S: 1.28	8000	*Staphylococcus aureus*	[185]
*Croton thurifer* Kunth.	0.07 ± 0.005	β-Elemene, germacrene D.	M: 35.39 S: 62.26	296.1	*Staphylococcus aureus*	[155]
				148	*Bacillus subtilis*	
				>1000	*Escherichia coli*	
				>1000	*Pseudomonas aeruginosa*	
				>1000	*Candida albicans*	
*Croton tricolor* Baill.	Nd	Epiglobulol, α-bisabolol, *trans*-α-bergamotol, β-caryophyllene, α-acorenol.	M: 3.4 S: 88.6	1.0 to 1024	*Candida* strains	[186]
*Croton urucurana* Baillon.(Leaves)	0.35	Bicyclogermacrene, germacrene D, germacrene D-4-ol, α-cadinol.	S: 85.9 Other: 2.8	10	*Staphylococcus aureus*	[6]
				10	*Staphylococcus epidermidis*	
				10	*Pseudomonas aeruginosa*	
				10	*Bacillus subtilis*	
				10	*Klebsiella pneumoniae*	
				10	*Escherichia coli*	
				10	*Salmonella setubal*	
				5	*Saccharomyces cerevisiae*	
				>20	*Candida albicans*	
*Croton urucurana* Baillon. (Stem bark)	0.05	Borneol, cadina-4,10(14)-dien-1α-ol, sesquicineole, bornyl acetate, γ-gurjunene epoxide.	M: 34 S: 57.3	2500	*Staphylococcus aureus*	[6], [187]
				1250	*Staphylococcus epidermidis*	
				2500	*Pseudomonas aeruginosa*	
				10,000	*Bacillus subtilis*	
				5000	*Klebsiella pneumoniae*	
				1250	*Escherichia coli*	
				2500	*Salmonella setubal*	
				5000	*Saccharomyces cerevisiae*	
				5000	*Cryptococcus neoformans*	
				10,000	*Candida albicans*	
*Croton zambesicus* Mull-Arg.	0.28	1,8-Cineole, cymene, α-terpineol, L-linalool.	M: 69.84 S: 15.62	16.0 250.0 16.0 16.0	*Escherichia coli* *Pseudomonas aeruginosa* *Bacillus subtilis* *Staphylococcus aureus*	[188]
*Croton zehntneri* Pax and K. Hoffm.		Estragole, (*E*)-anethole, bicyclogermacrene.		>5000	*Candida albicans*	[56]
				2500	*Candida tropicalis*	
				620–1250	*Microsporum canis*	
*Croton zehntneri* Pax and Hoffm.	Nd	Estragole, 1,8-cineole, eugenol.	M: 13.61 S: 1.7 P: 82.1	Synergistic effect with norfloxacin.	*Staphylococcus aureus*	[53]
	Bark	Estragole, (*E*)-anethole.	M: 1.92 S: 0.47 P: 95.96	Inhibition zone diameter (mm): 8.0 at 10 mL.	*Staphylococcus aureus*	[54]
				Inhibition zone diameter (mm): 19.3 at 10 mL.	*Candida parapsilosis*	
	Leaves	Estragole.	M: 0 S: 4.5 P: 93.94	Inhibition zone diameter (mm): 8.3 at 10 mL.	*Staphylococcus aureus*	[54]
				Inhibition zone diameter (mm): 19.0 at 10 mL.	*Candida parapsilosis*	
*Croton zehntneri* Pax and Hoffm.		Estragole, 1,8-cineol, eugenol.	M: 13.61 S: 1.7 P: 82.1	25	*Shigella fl exneri*	[15]
				Nd	*Salmonella typhimurium*	
				500	*Escherichia coli*	
				500	*Sthaphylococcus aureus*	
				500	*Streptococus β-haemolyticus*	
*Croton zehntneri* Pax and K. Hoffm. (Fresh leaves)	1.8	Estragole, spathulenol.	M: 85.0 S: 12.0	58.75	*Bacillus subtilis*	[14]
				63.15	*Bacillus megaterium*	
				145.0	*Staphylococcus aureus*	
				63.43	*Shigella sonnei*	
				38.52	*Salmonella paratyphi*	
				131.2	*Blastomyces dermatitidis*	
				58.75	*Candida albicans*	
				61.54	*Pityrosporum ovale*	
				88.51	*Cryptococcus neoformans*	

^a^: The order of the compounds is sorted by percentage from high to low and greater than 5.0%. Nd: Not defined; NI: not inhibited; M: monoterpenoids, S: sesquiterpenoids, B: benzenoids. P: phenylpropanoids.

## Data Availability

All data are available upon reasonable request from the corresponding authors (N.H.H. and W.N.S.).

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
