# Peer review of "Chemistry and Bioactivity of Croton Essential Oils: Literature Survey and Croton hirtus from Vietnam"

_molecules, 2023, doi:10.3390/molecules28052361_

Round 1

Reviewer 1 Report

Comments to the Authors

The manuscript entitled “Chemical composition, bioactivity of essential oils of Croton 2 species and update from Vietnam” has been reviewed. The topic is quite innovative and interesting, anyway the MS is acceptable for publication after major revision, mainly considering the component of bibliographic research work. The section on experimental activities on C. hirtus are more comprehensive.

Concerning the mini-review, there is no evidence for a critical assessment of the data, and it remains unclear to the reader what the really salient findings are. Some points are interesting, but Authors have to remember that a review MUST be a critical analysis of the data and do not simply provide a long list of (certainly interesting) facts. In my opinion of reviewer of experience, the review may serve as background material for suggestions of less explored topics in a field, or, on the contrary, as the overview after the conduction of a research project in a specific topic in which the authors have gained state of the art results. The MS is a large list of information, but without a discussion able to point up significative findings. The material and methods section (3.2. Reference Search) presents a poor description of the data mining process, the eligibility criteria and rejection criteria of the cited works, number of chosen and rejected works. The articles have been selected with a solid and clear methodology? If yes, what? Guidelines for authors clearly recommend PRISMA guidelines. I suggest adding the number of researchers involved in this activity, it should be useful to describe in deep which kind of sources have been selected (Only articles? Also, books and books chapters? Only articles in English were selected or also in other idioms?? Etc.)

Conclusions should incorporate also some new research trends.  

Additionally:

Line 42 – The abstract should be mentioned the number of eligible articles which have been selected for the review.

Line 105 – “C. hirtus” needs Italic style, please check the entire draft.

Line 134 – “A. aegypti” needs Italic style, please check the entire draft.

Line 141 – I suggest deleting the column “Mosquito” because only one species has been mentioned (Aedes aegypti). Maybe the title of the table can be changed to “Summary of Aedes aegypti larvicidal activity of essential oils of Croton spp.”

Reviewer 2 Report

The authors carried out a very complete review of the chemical and bioactivity background regarding species of the vast genus Croton. This update and what is read in this manuscript about the Croton hirtus species are appreciated.

I have reviewed an extensive manuscript (review), with 175 references, some of them cited up to 44 times and detected very few inconsistencies.

For example, in the introduction section, between lines 85 to 93 you will read interesting comments on the role of essential oils and yet, there is only one quote (2 authors). Due to the importance of the topic being investigated, I think it is necessary to review and quote at this point.

On page 10, Table 3, it is read in Sabinene, the citation (52) b (non-existent). Please revise.

Regarding the conclusions, it seems to me that the relevance of the research is not expressed in terms of its own results with the species C. hirtus, where only the contribution to its chemical and biological knowledge is noted. There could be comparative data that highlights its importance.

Reviewer 3 Report

I am attaching the article with numerous observations to review.

The method used to identify the components of the essential oil studied is totally unacceptable. Furthermore, several of them are relatively rare or almost unknown, so it is not possible to identify them as the authors propose. Both this and explaining what limits the review has to be characterized as a "mini" review, are elements that must be thoroughly reviewed.

The title should be modified, according to the results informed

Reviewer 4 Report

Dear, Huy Hung Nguyen, Ph.D.

Please check and revise.

-------------------------------------------------------------------------------------------------------------------------

Is this manuscript original paper? It looks like a review paper with too much reference data.

If you don't have much experimental data, why don't you submit it to letter-type?

-------------------------------------------------------------------------------------------------------------------------

Round 2

Reviewer 1 Report

Dear Authors,

the MS is now available for publication.

Kind regards,

Author Response

We sincerely thank the Reviewer for making our manuscript complete!

Reviewer 4 Report

Dear, Huy Hung Nguyen, Ph.D.

Please check and revise.

-------------------------------------------------------------------------------------------------------------------------

As you describe “This work has presents a literature survey “ in 4. Conclusions,  Is this manuscript a survey paper? I think survey papers are in the same category as review papers not original papers.

-------------------------------------------------------------------------------------------------------------------------
